

# N$_2$O isotope approaches for source partitioning of N$_2$O production and estimation of N2O reduction – validation with $^{15}$N gas-flux method in laboratory and field studies

Dominika Lewicka-Szczebak[1], Maciej Piotr Lewicki[2] and Reinhard Well[3]

[1] Centre for Stable Isotope Research and Analysis, University of Göttingen, Büsgenweg 2, 37077 Göttingen, Germany

[2] Institute of Theoretical Physics, Univeresity of Wrocław, pl. M. Borna 9, 50-204 Wrocław, Poland

[3] Thünen-Institut of Climate-Smart Agriculture, Bundesallee 50, 38116 Braunschweig, Germany

*Correspondence to*: Dominika Lewicka-Szczebak (dominika.lewicka@uni-goettingen.de)

**Abstract.**

The approaches based on natural abundance N$_2$O stable isotopes are often applied for the estimation of mixing proportions between various N$_2$O producing pathways as well as for estimation of the extent of N$_2$O reduction to N$_2$. But such applications are associated with numerous uncertainties and hence their limited accuracy needs to be considered . Here we present the first systematic validation of these methods for laboratory and field studies applying the $^{15}$N gas-flux method as the reference approach.

Besides applying dual isotope plots for interpretation of N$_2$O isotopic data, for the first time we propose a three dimensional N$_2$O isotopocule model based on Bayesian statistics to estimate the N$_2$O mixing proportions and reduction extent based simultaneously on three N$_2$O isotopic signatures ($\delta^{15}$N, $\delta^{15}$N$^{SP}$ and $\delta^{18}$O). Determination of mixing proportions of individual pathways with N$_2$O isotopic approaches appears often imprecise, mainly due to imperfect isotopic separation of the particular pathways. Nevertheless, the estimation of N$_2$O reduction is much more robust, when applying optimal calculation strategy, reaching typically accuracy of N$_2$O residual fraction determination of about 0.1.



## 1. Introduction

Nitrous oxide ($N_2O$) emission from soils and waters may result from numerous nitrogen transformation processes, mainly heterotrophic bacterial denitrification (bD), autotrophic nitrification (Ni), nitrifier denitrification (nD), and fungal denitrification (fD), but also heterotrophic nitrification, chemodenitrification, or co-denitrification (Butterbach-Bahl et al., 2013; Firestone and Davidson, 1989; Müller et al., 2014). The ability to distinguish the proportional contributions of these various $N_2O$ origins ($f_{bD}$, $f_{Ni}$, $f_{nD}$, $f_{fD}$) is important in

constraining the N budget and in developing and assessing the performance of mitigation strategies for $N_2O$ emission, which significantly contributes to global warming and stratospheric ozone depletion (IPCC, 2007; Ravishankara et al., 2009). Partition of the mixing proportions $f_{bD}$, $f_{Ni}$, and $f_{nD}$ is only partially possible by combination of numerous experimental techniques, including sophisticated $^{15}N$ and $^{18}O$ isotope labelling techniques (Müller et al., 2014; Wrage-Mönnig et al., 2018). However, also natural abundance $N_2O$ isotopic

analyses have been often applied to estimate the possible proportional contribution of particular pathways (Toyoda et al., 2017; Yu et al., 2020) and are currently the only isotopic approach to identify $f_{fD}$ (Rohe et al., 2017; Wrage-Mönnig et al., 2018). The partition of mixing proportions based on natural abundance $N_2O$ isotopes is theoretically possible thanks to characteristic isotopic fractionation for each pathway, determined in numerous laboratory pure culture experiments (Toyoda et al., 2017), but practically very complex, mainly due to changes

of $N_2O$ isotopic signature during its partial reduction to $N_2$ and due to overlapping isotopic endmember values of individual pathways. $N_2O$ isotopic analyses comprise the isotopic determination of: oxygen ($\delta^{18}O$), bulk nitrogen ($\delta^{15}N$) and nitrogen site preference ($\delta^{15}N^{SP}$), i.e., the difference in $\delta^{15}N$ between the central and the peripheral N atom of the linear $N_2O$ molecules (Brenninkmeijer and Röckmann, 1999; Toyoda and Yoshida, 1999). All these three isotopic signatures ($\delta^{18}O$, $\delta^{15}N$ and $\delta^{15}N^{SP}$) show characteristic ranges of isotopic signatures for particular

$N_2O$ production pathway but are also altered during the $N_2O$ reduction process.

$N_2O$ reduction to $N_2$ occurs during the last step of microbial denitrification, i.e., anoxic reduction of nitrate ($NO_3^-$) to $N_2$ through the following intermediates: $NO_3^- \rightarrow NO_2^- \rightarrow NO \rightarrow N_2O \rightarrow N_2$ (Firestone and Davidson, 1989; Knowles, 1982). Commonly applied experimental techniques enable us to quantitatively analyse only the intermediate product of this process, $N_2O$, but not the final product, $N_2$ (Groffman, 2012; Groffman et al., 2006).

This is due to the high atmospheric $N_2$ background precluding direct measurements of $N_2$ emissions in presence of the natural atmosphere (Bouwman et al., 2013; Saggar et al., 2013). Estimation of $N_2$-flux is possible with sophisticated laboratory experiments applying $N_2$-free helium atmosphere (Scholefield et al., 1997) or $^{15}N$ gas-flux method, i.e. $^{15}N$ analyses of gas fluxes after addition of $^{15}N$-labelled substrate (Bergsma et al., 2001; Schmidt et al., 1998). Previous studies documented large possible variations in $N_2$ flux, and consequently also in

the residual unreduced $N_2O$ fraction: $r_{N2O} = y_{N2O}/(y_{N2}+y_{N2O})$ ($y$: mole fraction). In laboratory studies, the whole scale of possible $r_{N2O}$ variations, ranging from 0 to 1, had been found (Lewicka-Szczebak et al., 2017; Lewicka-



Szczebak et al., 2015; Mathieu et al., 2006; Morse and Bernhardt, 2013; Senbayram et al., 2012). Due to technical limitations, so far only the $^{15}$N gas-flux method had been applied in field conditions to determine $r_{N2O}$ (Aulakh et al., 1991; Baily et al., 2012; Bergsma et al., 2001; Buchen et al., 2016; Decock and Six, 2013;

Kulkarni et al., 2013; Mosier et al., 1986). Moreover, first attempt to apply the $^{15}$N gas-flux method under $N_2$-reduced atmosphere in field has been presented recently (Well et al., 2019a). This new approach increases the sensitivity of $^{15}$N gas-flux method which was so far very limiting for successful application in field studies (Buchen et al., 2016). But still, application of this approach is technically very demanding and applicable only with a low temporal and spatial resolution. Hence, no comprehensive data sets from field-based measurements of

soil $N_2$ emissions are available and this important component in soil nitrogen budget is still missing. This constitutes a serious shortcoming in understanding and mitigating the microbial consumption of nitrogen fertilisers (Bouwman et al., 2013; Seitzinger, 2008), and the $N_2O$ emission.

An alternative approach for assessing $N_2$ fluxes is the use of $N_2O$ isotopes, which allows to indirectly determine $r_{N2O}$ from its isotopic signature (Ostrom et al., 2007; Well and Flessa, 2009), since the magnitude of the observed

isotope effect due to $N_2O$ reduction depends largely on $r_{N2O}$ (Jinuntuya-Nortman et al., 2008; Menyailo and Hungate, 2006; Ostrom et al., 2007; Well and Flessa, 2009). This approach is also potentially applicable for quantification of $r_{N2O}$ in field conditions (Buchen et al., 2018; Park et al., 2011; Toyoda et al., 2011; Verhoeven et al., 2019; Zou et al., 2014). Its advantage over the $^{15}$N gas-flux method lies in its easier and non-invasive application, no need of additional fertilization, and much lower costs. But on the other hand, complexity of the

$N_2O$ production pathways with co-occurring $N_2O$ reduction and variability of isotope effects can make this estimation imprecise (Wu et al., 2019). Since two processes, mixing and reduction, determine the final $N_2O$ isotopic signature, we need at least two isotopic values to be able to asses both: $N_2O$ mixing ratio between two $N_2O$ production pathways and $r_{N2O}$. Therefore, often applied are the dual isotope plots, also called isotope Mapping approach (Map), *i.e.*, isotopic relations in the space $\delta^{15}N^{SP}/\delta^{15}N$ (SP/N Map) and $\delta^{15}N^{SP}/\delta^{18}O$ (SP/O

Map). The SP/N Map has been first applied for agricultural soils by Toyoda et al. (2011). Afterwards many studies utilized this relation to determine $N_2O$ mixing proportions and $N_2O$ reduction (Kato et al., 2013; Wolf et al., 2015; Zou et al., 2014). Later, it was shown that $\delta^{18}O$ can be also used as a good tracer for $N_2O$ production processes, thanks to high O-exchange during bD resulting in quite stable $\delta^{18}O$ values for this pathway (Lewicka-Szczebak et al., 2016). Based on this finding the SP/O Map for $N_2O$ interpretation was proposed (Lewicka-

Szczebak et al., 2017) and applied in recent studies (Buchen et al., 2018; Ibraim et al., 2019; Verhoeven et al., 2019; Wu et al., 2019). Both SP/N and SP/O Map have been applied jointly for field studies (Ibraim et al., 2019) and showed quite a good agreement in the calculated $r_{N2O}$ and $f_{bD}$ values. However, so far these two approaches were not combined together into a complex three-dimensional model allowing the calculation of pathways mixing proportions and $r_{N2O}$ based on three isotopic signatures ($\delta^{15}N$, $\delta^{18}O$, $\delta^{15}N^{SP}$) simultaneously.

Development of such a model is a clear current need.





Precise quantification of both, the production pathway proportions and the extent of $N_2O$ reduction with isotope Maps is limited by wide ranges of isotopic signatures reported for individual pathways, the overlapping of these isotopic signatures ranges, variations in substrate isotopic compositions, and variability of fractionation factors associated with $N_2O$ reduction (Toyoda et al., 2017; Yu et al., 2020). Hence, it can be questioned how far we can

trust the quantitative results provided by calculations based on isotope Maps. To answer this question comparisons with estimates based on independent methods are needed. The first attempt for comparing $r_{N2O}$ obtained with SP/O Map and $^{15}N$ gas-flux method in a field case study was performed by Buchen et al. (2018). Due to non-identical treatment and differences in soil moisture and mineral N, the results of both treatments were difficult to compare, however, the $r_{N2O}$ values obtained indicated clearly the dominance of $N_2$ flux over

$N_2O$ flux by both methods. That study also presented analysis of various calculation scenarios applying upper and lower limits for mixing isotopic endmembers values and reduction fractionation factors, which revealed pronounced uncertainty of this calculation approach (Buchen et al., 2018). It was suggested that a further study on validation and uncertainty analysis of the SP/O Map is required with particular attention to identical treatment for both approaches under comparison. Another comparison was performed with archival datasets applying

helium incubations as reference method and indicated large uncertainties of the calculations based on the SP/O Map (Wu et al., 2019). The huge uncertainties determined in these studies resulted from the fact that the full range of endmember values and fractionation factors reported in the literature was taken into account. But for particular soils and experimental conditions these ranges might be smaller and uncertainties thus lower. Hence, it is still unsure to which extent the ranges of isotopic fractionation factors determined in laboratory conditions and

for pure culture studies are valid for particular experiments. It is not feasible to validate each isotope characteristic separately in field studies, since the pathways are not easily separable and this can be only achieved in controlled laboratory conditions.

While these recent studies indicated severe imprecision associated with the $r_{N2O}$ estimations based on $N_2O$ isotopocule approaches (Buchen et al., 2018; Wu et al., 2019), the suitability of this approach in estimation of

$r_{N2O}$ and mixing proportions has never been validated in a systematic study with a reference method. Hence, the idea of this study is to validate the methods based on $N_2O$ isotope Maps and determine their attainable precision by parallel application with the reference method. We compare the calculated $N_2$ flux based on the $^{15}N$ gas-flux method ($^{15}N$ treatment) and $N_2O$ isotope Maps (natural abundance (NA) treatment) in laboratory and field experiments applying identical treatment strategy. Moreover, we present a new three-dimensional isotopocule

model (3DI model) based on 3D isotopocule space and provide a validation of its outputs. This is the first attempt to systematically validate the results from $N_2O$ natural abundance isotopic studies ($N_2O$ isotopocule approaches) in laboratory and field conditions.



Our aim is to (1) validate applicability of $N_2O$ isotopocule approaches for $N_2$ flux determination, (2) validate applicability of $N_2O$ isotopocule approaches for partition of $N_2O$ producing pathways and (3) to develop best evaluation strategy for interpretation of $N_2O$ isotopic data.

## 2 Methods

### 2.1 Field study

Silt loam soil *Albic Luvisol* from arable cropland of Merklingsen experimental station located near Soest (North Rhine-Westphalia, Germany, 51°34'15.5"N, 8°00'06.8"E) was used (87% silt, 11% clay, 2% sand). The soil density of intact cores was 1.3 g ml$^{-1}$, pH value 6.8, total C content 1.30%, total N content 0.16%, organic matter content 2.14%. The field was sown with winter rye in September 2015 and mineral under foot fertilization was applied. Our experiments were conducted on experimental plots of a field study on management effects on greenhouse gas fluxes. We selected the 'climate-optimized farm' treatment where a complex cropping rotation of silage maize - winter wheat - faba bean – winter barley – perennial rye had been established since 2010 (Kramps-Alpmann et al., 2017). This treatment was managed by zero-tillage with direct seeding and fertilisation was a combination of organic (biogas digestate) and mineral fertilizer where doses were set according to official fertilizer recommendations (Baumgärtel and Benke, 2009). On 13 October in each of the four replicate plots (6 * 12 m) we established microplots consisting of aluminum cylinders (length 35cm, diameter 15cm) inserted to 30cm depth into the soil so that 5cm extended above the ground for installation of the flux chamber. Three field campaigns were carried out in November 2015 (F1), March 2016 (F2) and Mai/June 2016 (F3). After each field campaign the cylinders were removed, cleaned and later reinstalled on new locations for the next field campaign (on 27 Nov 2015 for F2 sampling and on 28 April 2016 for F3 sampling).

On each replicate plot cylinders were installed pairwise – one for gas flux measurements and one for mineral nitrogen sampling – for 3 treatments – natural abundance (NA), traced nitrate ($^{15}NO_3^-$) and traced ammonium ($^{15}NH_4^+$) – in total 6 cylinders per replicate plot. The distance between each treatment cylinder was at least 2m, pair of cylinders for one treatment were in 0.5m distance.

At the beginning of the experiment, a fertilizer solution with 240 mg N L$^{-1}$ as NaNO$_3$ and 240 mg N L$^{-1}$ as NH$_4$Cl was added to the experimental microplots through needle injection technique. Three mL of the fertilizer solution was injected into 72 points using 12 needles inserted subsequently into 6 depths (2.5 - 7.5 - 12.5 - 17.5 - 22.5 - 27.5 cm) from the top to the bottom using peristaltic pump. This strategy was based on previous studies (Buchen et al., 2016; Wu et al., 2011) and was enhanced by pre-experimental tests to obtain the most homogeneous tracer distribution (Lewicka-Szczebak and Well, 2020). Total fertilization was 10 mg N per kg soil which was equivalent to about 40 kg N per ha.





In total, 216 mL of fertilizing solution was inserted into each microplot which resulted in 3 % increase in water
content. For $^{15}$N-labelled treatments the $^{15}$N content in fertilizing solution was calculated to achieve about 60
atom % $^{15}$N in the $^{15}$N-labelled N pool. The $^{15}$NO$_3^-$ treatment received tracer solution containing  68 atom % $^{15}$N
and the $^{15}$NH$_4^+$ treatment received 64 atom % $^{15}$N.

Immediately after fertilizing solution addition, the flux chamber microplots were closed for gas accumulation.
Opaque PVC chambers of an area of 1.767 dm$^2$ and a volume of 2.65 dm$^3$ were applied with installed valves for
sample collection and a fan for gas mixing. The closed chamber method (Hutchinson and Mosier, 1981) was
used for N$_2$O flux measurement. Chambers were closed and sealed with air-tight rubber bands for 120 min and
headspace sampling was performed after 40, 80 and 120 min into evacuated crimped 20 mL vials with a 30 mL
syringe for gas-flux measurements. Additionally, after 120 min, samples for isotope analysis were collected. For
$^{15}$N treatments two identical replicates were taken into 12 mL evacuated screw-cup Exetainers® (Labco Limited,
Ceredigion, UK) with two combined 15 mL syringes. For the NA treatment, one gas sample was transferred into
an evacuated 115 mL crimp-cap vial with a 150 mL syringe.

Each field campaign lasted 5 days. Gas samples were collected once on the first day after fertilization, afterwards
twice a day – in the morning and in the evening, and once on the last 5$^{th}$ day in the morning.

The soil sampling microplots were treated identically and used for mineral nitrogen sampling. The soil samples
were collected with a Goettinger boring rod with 18 mm outer diameter and 14 mm slots (Nietfeld GmbH,
Quakenbrück, Germany). Boreholes were sealed by inserting a closed sand-filled PVC pipe with the same
diameter as the bore. For each sampling, three cores were collected and homogenised to one mixed sample each
day, hence we performed 5 soil samplings during each campaign. The samples were immediately transported to
the laboratory at 6°C and mineral nitrogen extractions were performed on the same day.

**2.2 Laboratory incubation**

The soil from the experimental field site was used to prepare incubation columns for laboratory incubation. The
soil was air dried and sieved at 4 mm mesh size. Afterwards, the soil was rewetted to achieve  a water content
equivalent to 60 % water-filled pore space (WFPS) and fertilised with 20 mg N per kg soil, added as NaNO$_3$ (10
mg N) and NH$_4$Cl (10 mg N). Analogically as in the field study, three treatments were prepared: natural
abundance (NA), labelled with $^{15}$N nitrate ($^{15}$NO$_3$) and labelled with $^{15}$N ammonium ($^{15}$NH$_4$). For the $^{15}$NO$_3$
treatment, NaNO$_3$ solution with 72 atom % $^{15}$N was added and for the $^{15}$NH$_4$ treatment, NH$_4$Cl solution with 63
atom % $^{15}$N was added. Then soils were thoroughly mixed to obtain homogenous distribution of water and
fertilizer and an equivalent of 1.69 kg dry soil was repacked into each incubation column with bulk density of
1.3 g cm$^{-3}$.



For each treatment 14 columns were prepared, and half of them received additional water injected on the top of the column (100 mL water added) to prepare two moisture treatments: dry (61 % WFPS) and wet (72 % WFPS). The incubation lasted 12 days. In the meantime, on the 6[th] day of incubation, water addition on the top of each column was repeated (80 mL water added) to increase the soil moisture in both treatments to ca. 68 % WFPS in the dry treatment and ca. 81 % WFPS in the wet treatment. The strategy of adding water on the top of the

column to achieve target water content was necessary to allow mixing and compaction at a suitable (low) water content of the soil and thus to optimise homogeneity of water and fertilizer distribution (Lewicka-Szczebak and Well, 2020). The incubation temperature was 20°C. The columns were continuously flushed with a gas mixture with reduced $N_2$ content to increase the measurements sensitivity (2% $N_2$ and 21% $O_2$ in He, (Lewicka-Szczebak et al., 2017)) with a flow of 9 mL min[-1]. Gas samples were collected daily into two 12 mL septum-capped

Exetainers® (Labco Limited, Ceredigion, UK) and one crimped 100 mL vial connected to the vents of the incubation columns. Soil samples were collected 5 times during the incubation by sacrificing one incubation column per sampling event, which was then divided into three subsamples (replicate samples of mixed soil).

**2.3 Gas analyses**

Measurements of $N_2O$ concentrations in the 20 mL samples were carried out with a gas chromatograph (GC,

2014; Shimadzu, Duisburg, Germany) equipped with an electron capture detector (ECD) and an autosampler (Loftfields Analytical Solutions, Neu Eichenberg, Germany). The analytical precision was around 2%.

Flux rates of total $N_2O$ for field campaigns, *i.e.*, including fluxes from [15]N-labelled and non-labelled sources, were calculated from ordinary linear regression of the four consecutive samples over time using the R package gasfluxes (Fuß, 2015) and the following equation:

$$J_{N2O} = \frac{dc_{N2O}}{dt} * \frac{V}{A} \qquad (1)$$

where $J_{N2O}$ is the flux rate in µg $N_2O$-N m[-2] h[-1], $C_{N2O}$ is $N_2O$ mass concentration in µg N m[-3] corrected by the chamber temperature according to the ideal gas law, *t* is closing time of the chamber, *V* is volume of the chamber in m[3] and *A* is covered soil area in m[2].

For laboratory incubations due to constant flow-through the following equation was applied:

$$J_{N2O} = C_{N2O} * \frac{Q}{A} \qquad (2)$$

where $J_{N2O}$ is the flux rate in µg $N_2O$-N m[-2] h[-1], *C* is $N_2O$ mass concentration in µg N m[-3] corrected by the incubation temperature according to the ideal gas law, *Q* is the gas flow rate through the incubation vessels in m[3] h[-1], and *A* is soil area in the incubation vessel in m[2].

The gas samples collected from [15]N treatments were analyzed for [15]N content with a modified GasBench II preparation system coupled to MAT 253 isotope ratio mass spectrometer (Thermo Scientific, Bremen, Germany)



according to Lewicka-Szczebak et al. (2013). In this set-up, $N_2O$ is converted to $N_2$ prior to analysis, which allows simultaneous measurement of stable isotope ratios $^{29}R$ ($^{29}N_2/^{28}N_2$) and $^{30}R$ ($^{30}N_2/^{29}N_2$), of $N_2$, of the sum of denitrification products ($N_2+N_2O$) and of $N_2O$. Based on these measurements the following values are

calculated according to the respective equations (after Spott et al. (2006)):

The $^{15}N$ abundance of $^{15}N$-labelled pool ($a_P$) from which $N_2$ ($a_{P\_N2}$) or $N_2O$ ($a_{P\_N2O}$) originate is calculated as follows:

$$a_P = \frac{^{30}x_M - a_M \cdot a_{bgd}}{a_M - a_{bgd}} \qquad (3)$$

The calculation of $a_P$ is based on the non-random distribution of $N_2$ and $N_2O$ isotopologues (Spott et al., 2006)

where $^{30}x_M$ is the fraction of $^{30}N_2$ in the total gas mixture:

$$^{30}x_M = \frac{^{30}R}{1 + ^{29}R + ^{30}R} \qquad (4)$$

$a_M$ is $^{15}N$ abundance in total gas mixture

$$a_M = \frac{^{29}R + 2 \; ^{30}R}{2(1 + ^{29}R + ^{30}R)} \qquad (5)$$

$a_{bgd}$ is $^{15}N$ abundance of non-labelled pool (atmospheric background or experimental matrix)

The fraction originating from the $^{15}N$-labelled pool ($f_P$) for $N_2$ ($f_{P\_N2}$), $N_2+N_2O$ ($f_{P\_N2+N2O}$) and $N_2O$ ($f_{P\_N2O}$) within the total N of the sample is calculated as follows:

$$f_P = \frac{a_M - a_{bgd}}{a_P - a_{bgd}} \qquad (6)$$

The fraction originating from the $^{15}N$-labelled pool within the sample ($f_{N2}$) is calculated, taking into account the actual $N_2$ concentration background in the sample $C_{N2}$:

$$f_{N2} = f_{P\_N2} * C_{N2} \qquad (7)$$

From the $f_{N2}$ value determined with Eq.7 the $N_2$ flux was calculated, in the same manner as for $N_2O$, for field campaigns (Eq. 1):

$$J_{N2} = \frac{f_{N2}}{dt} * \frac{V}{A} \qquad (8)$$

where $J_{N2}$ is the $N_2$ flux rate in µg $N_2$-N m$^2$ h$^{-1}$, $f_{N2}$ is $N_2$ mass concentration in µg N m$^3$ corrected by the chamber

temperature according to the ideal gas law, $t$ is closing time of the chamber, $V$ is volume of the chamber in m$^3$ and $A$ is covered soil area in m$^2$. Chamber closing time was 120 min and for one chosen field study (F3) the linearity of $N_2$ increase over 120 min was checked and confirmed. The fluxes correction for underestimation due to subsoil flux and gas soil storage (Well et al., 2019b) was not performed because the focus of this paper was to determine $r_{N2O}$ while subsoil diffusion of $N_2$ and $N_2O$ is almost identical. This correction would thus not





significantly impact $r_{N2O}$. But the fluxes shown in Fig. S2 are measured fluxes and include the underestimation of [15]N-based estimates (Well et al., 2019b).

For laboratory incubations with the constant flow through $N_2$ flux was determined in the same manner as respectively for $N_2O$ (Eq. 2):

$$J_{N2} = f_{N2} * \frac{Q}{A} \tag{9}$$

where $J_{N2}$ is the $N_2$ flux rate in µg $N_2$-N m$^{-2}$ h$^{-1}$, $f_{P\_N2}$ is $N_2$ mass concentration in µg N m$^3$ corrected by the chamber temperature according to the ideal gas law, $Q$ is the gas flow rate through the incubation vessels in m$^3$ h$^{-1}$, and $A$ is soil area in the incubation vessel in m$^2$.

$N_2O$ residual fraction ($r_{N2O}$) representing the unreduced $N_2O$ mole fraction of total gross $N_2O$ production
(Lewicka-Szczebak et al., 2017) is calculated as:

$$r_{N2O} = \frac{J_{N2O}}{J_{N2O}+J_{N2}} \tag{10}$$

where $J_{N2O}$ and $J_{N2}$ are the $N_2O$ and $N_2$ flux rates in µg $N_2O$-N m$^{-2}$ h$^{-1}$.

The analytical detection limit of the calculated $N_2$ flux from the [15]N labelled pool was approx. 50 µg N m$^2$ h$^{-1}$ for field studies and approx. 1.5 µg N m$^2$ h$^{-1}$ for laboratory experiments (due to increased sensitivity as a result of
the $N_2$-reduced atmosphere).

The gas samples collected in NA treatments were analyzed for isotopocule $N_2O$ signatures using a Delta V isotope ratio mass spectrometer (Thermo Scientific, Bremen, Germany), coupled to an automatic preparation system with Precon + Trace GC Isolink (Thermo Scientific), where $N_2O$ was pre-concentrated, separated and purified and m/z 44, 45, and 46 of the intact $N_2O^+$ ions as well as m/z 30 and 31 of $NO^+$ fragment ions were
determined. The results were evaluated accordingly (Röckmann et al., 2003; Toyoda and Yoshida, 1999; Westley et al., 2007) which allows the determination of average $\delta^{15}N$, $\delta^{15}N^\alpha$ ($\delta^{15}N$ of the central N position of the $N_2O$ molecule), and $\delta^{18}O$. $\delta^{15}N^\beta$ ($\delta^{15}N$ of the peripheral N position of the $N_2O$ molecule) was calculated as $\delta^{15}N$ =($\delta^{15}N^\alpha + \delta^{15}N^\beta$)/2 and [15]N site preference ($\delta^{15}N^{SP}$) as $\delta^{15}N^{SP} = \delta^{15}N^\alpha - \delta^{15}N^\beta$.

Pure $N_2O$ analysed for isotopocule values in the laboratory of the Tokyo Institute of Technology was used as
internal reference gas applying calibration procedures reported previously (Toyoda and Yoshida, 1999; Westley et al., 2007). Moreover, the standards from a laboratory inter-comparison (REF1, REF2) were used for performing two-point calibration for $\delta^{15}N^{SP}$ values (Mohn et al., 2014). All isotopic values are expressed as ‰ deviation from the [15]N/[14]N and [18]O/[16]O ratios of the reference materials (i.e. atmospheric $N_2$ and Vienna Standard Mean Ocean Water (VSMOW), respectively). The analytical precision determined as standard
deviation (1σ) of the internal standards for measurements of $\delta^{15}N$, $\delta^{18}O$, and $\delta^{15}N^{SP}$ was typically 0.1, 0.1, and 0.5 ‰, respectively.



### 2.4 Soil analyses

All soil samples were homogenized. Soil water content was determined by weight loss after 24 h drying in 110ºC. Soil pH was determined in 0.01 mol $CaCl_2$ solution (ratio 1:5). Nitrate and ammonium concentration was

determined by extraction in 2M KCl in 1:4 ratio by 1h shaking. Nitrite concentration was determined in alkaline extraction solution of 2M KCl with addition of 2M KOH (25 mL per L) in 1:1 ratio for 1 minute of intensive shaking (Stevens and Laughlin, 1995). The amount of added KOH was adjusted to keep the alkaline conditions in extracts (pH over 8). After shaking, the samples were centrifuged for 5 minutes and filtrated. The extracts for $NO_2^-$ measurements were stored at -4 °C and analyzed within 5 days. $NO_3^-$, $NH_4^+$ and $NO_2^-$ concentrations were

determined colorimetrically with an automated analyser (Skalar Analytical B.V., Breda, the Netherlands).

To determine isotopic signatures of mineral nitrogen in NA treatments, microbial analytical methods were applied. For nitrate, the bacterial denitrification method with *Pseudomonas aureofaciens* was applied (Casciotti et al., 2002; Sigman et al., 2001)). For nitrite, the bacterial denitrification method for selective nitrite reduction with *Stenotrophomonas nitritireducens* was applied (Böhlke et al., 2007), also for $^{15}N$-enriched samples from

$^{15}N$ treatments. For ammonium, a chemical conversion to nitrite with hypobromite oxidation (Zhang et al., 2007) followed by bacterial conversion of nitrite after pH adjustment was applied (Felix et al., 2013).

In $^{15}N$ treatments, $^{15}N$ abundances of $NO_3^-$ ($a_{NO3-}$) and $NH_4^+$ ($a_{NH4+}$) were measured according to the procedure described in Stange et al. (2007) and Eschenbach et al. (2017). $NO_3^-$ was reduced to NO by Vanadium-III chloride ($VCl_3$) and $NH_4^+$ was oxidized to $N_2$ by hypobromite ($NaOBr$). NO and $N_2$ were used as measurement

gas. Measurements were performed with a quadrupole mass spectrometer (GAM 200, InProcess, Bremen, Germany).

### 2.5  $N_2O$ isotope mapping approach (Map)

The Mapping approach is based on the different slopes of the mixing line between bD and fD or Ni and the reduction line reflecting isotopic enrichment of residual $N_2O$ due to its partial reduction in dual isotope plots.

Both lines are defined from the known most relevant literature data on the respective mixing endmembers isotopic signatures and reduction fractionation factors. The detailed isotopic characteristics applied for the isotope Maps are presented in the supplement (Table S1) and follow the most recent review paper (Yu et al., 2020). The detailed calculation strategy for SP/O Map can be found in the Supplement for the Wu et al. (2019) paper and for SP/N Map in the Supplement for the Toyoda et al. (2011) paper. The calculations are performed

according to two possible cases of $N_2O$ mixing and reduction:

- Case 1 - $N_2O$ produced from bD is first partially reduced to $N_2$, followed by mixing of the residual $N_2O$ with $N_2O$ from other pathways,
- Case 2 - $N_2O$ produced by various pathways is first mixed and afterwards reduced.



The calculations can be performed following different scenarios of particular endmember mixing: either bD-fD
mixing or bD-Ni mixing. For our case studies, we rather expect higher fD contribution than Ni, hence the bD-fD
mixing was applied and contribution of Ni was neglected. In the supplement, we also present a comparison of
calculation results based on both mixing scenarios bD-fD and bD-Ni (Table S2 and supplementary spreadsheet
table).

### 2.6  Three-dimensional N₂O isotopocule model (3DI model)

The probability distributions of proportional contributions _f_i_ were determined using a stable isotope mixing
model in the Bayesian framework (Parnell et al., 2013). This allowed us to integrate three $N_2O$ isotopic
signatures into one model to find the nearest solution for the $r_{N2O}$ and mixing proportions.

The core of the model was based on the work of Moore and Semmens (2008) which was further extended with
implementation of N₂O reduction in two possible cases (analogically as for Map – see Section 2.5):

Case 1)        $f_{bD}\left(\delta_{bD} + \varepsilon\, ln(r_{bD})\right) + f_{nD}\delta_{nD} + f_{fD}\delta_{fD} + f_{Ni}\delta_{Ni} = \delta_{N2O}$                    (11)

Case 2)        $f_{bD}\delta_{bD} + f_{nD}\delta_{nD} + f_{fD}\delta_{fD} + f_{Ni}\delta_{Ni} + \varepsilon\, ln(r_{N2O}) = \delta_{N2O}$                    (12)

where f stands for fraction of N₂O originating from a particular pathway and δ stands for isotopic signature
characteristic of this pathway, respectively for bD, nD, fD and nitrification Ni. ε is the isotope fractionation
factor for N₂O reduction to N₂ and $r_{N2O}$ is the N₂O residual fraction as defined in Eq. 10. $r_{bD}$ is the N₂O residual
fraction of bacterial denitrification only, as it is assumed in Case 1. This value can be recalculated to obtain $r_{N2O}$
as follows:

$$r_{N2O} = f_{bD}r_{bD} + f_{nD} + f_{fD} + f_{Ni}$$                    (13)

Let us briefly summarize the key assumptions and features of the statistical model. The input data of measured _m_
isotope signatures (here three: $\delta^{15}N$, $\delta^{15}N^{SP}$, $\delta^{18}O$) from _n_ sources (here four: bD, nD, fD and Ni) is assumed to
be normally distributed and multiple measurements (here: 1 to 7 replicates) constitute a single sample, on which
the Monte-Carlo integration is performed. The uncertainties of the source's data is fed into the model through the
variance in the calculation of unnormalized likelihood (see eq. 16). Prior distributions of parameters were
assumed uninformative, _i.e.,_ flat Dirichlet distribution was used for proportional source contributions _f_i_ and
uniform distribution for reduction parameter _r_. For each random sample _(f_i, r)_ a mean and a variance of each
isotope signature _j_ are calculated (different for two cases listed above):

Case 1)        $\mu_j = \sum_{i=1}^{n}(f_i\delta_{ij}) + f_{bD}\,\varepsilon\, ln(r_{bD}), \sigma_j = \sqrt{\sum_{i=1}^{n}(f_i\sigma_{ij}^2) + f_{bD}|ln(r_{bD})|\sigma_{\varepsilon j}^2}$                    (14)

Case 2)        $\mu_j = \sum_{i=1}^{n}(f_i\delta_{ij}) + \varepsilon\, ln(r_{N2O}), \sigma_j = \sqrt{\sum_{i=1}^{n}(f_i\sigma_{ij}^2) + |ln(r_{N2O})|\sigma_{\varepsilon j}^2}$                    (15)

and the likelihood of such a combination is calculated as:




$$L(x \mid \mu_j, \sigma_j) = \prod_k^N \prod_j^m \left[ \frac{1}{\sigma_j \sqrt{2\pi}} exp \left( \frac{-(x_{kj} - \mu_j)^2}{2\sigma_j^2} \right) \right] \tag{16}$$

where $x\_kj$ stands for $k$-th measurement of the sample and $j$-th isotope signature. We use the Markov-chain Monte-Carlo with the Metropolis condition: $L\_\{i+1\}/L\_\{i\}c >= $ alpha, where alpha is a random variable sampled from a uniform distribution.

The detailed input parameters for the model are presented in the supplement (Table S1). The detailed isotopic characteristics to be applied for the isotope signatures of mixing endmembers and reduction fractionation factors

are adopted after the most recent review paper (Yu et al., 2020).

### 2.7 Statistics

For results comparisons, an analysis of variance was used with the significance level α of 0.05. The uncertainty values provided for the measured parameters represent the standard deviation ($1\sigma$) of the replicates. The propagated uncertainty was calculated using Gauss' error propagation equation taking into account standard

deviations of all individual parameters.

The agreement with the reference method was assessed with the Nash–Sutcliffe efficiency ($F$) (Nash and Sutcliffe, 1970), which represent the $R$ of the fit to the 1:1 line between observed reference ($O$) and estimated ($E$) values, as also used in previous validation studies (Lewicka-Szczebak et al., 2017; Wu et al., 2019):

$$F = 1 - \frac{\sum_{i=1}^n (O_i - E_i)^2}{\sum_{i=1}^n (O_i - O)^2} \tag{17}$$

where $E_i$ is the $r_{N2O}$ value estimated with the method under validation, corresponding to the observed $r_{N2O}$ value determined with the reference method: $O_i$, and $O$ is the observed mean. In this assessment, an $F=1$ refers to a perfect fit between estimated and reference values, lower $F$ values indicate worse model fits, whereas a negative $F$ occurs when the observed mean is a better predictor than the model.

### 3. Results

### 3.1 Soil properties

Soil organic N was analyzed in soil samples from each sampling campaign and varied only slightly with content of 0.141 ± 0.007 % N and isotopic signature $\delta^{15}N$ of 7.4 ± 0.4‰. $\delta^{18}O$ of soil water varied only slightly for field campaigns and equaled -6.7 ‰ for F1, -7.0 ‰ for F2, and -6.4 ‰ for F3, but was higher for incubation experiments with mean of -5.3 ‰. Detailed characteristics for mineral nitrogen contents and isotopic signatures

are presented in Table 1. The variations in water and nitrate content during the field campaigns and laboratory incubations with comparison between NA and $^{15}N$ treatment are presented in the supplement (Fig. S1). Importantly, for vast majority of sampling points these soil conditions are well comparable between both





treatments which allows for the methods comparison. Significant difference was only noted for nitrate content for the last sample in L2 and for water content for the last sample in F1 (Fig. S1).

### 3.2 Field campaigns

The first field campaign F1 in Nov 2015 (23$^{rd}$ Nov-27$^{th}$ Nov) showed low $N_2O$ fluxes from 1.2 to 33.2 g N-$N_2O$ ha$^{-1}$ d$^{-1}$ (Table 1). $N_2O$ isotopic signatures were determined for all the samples except one. The $N_2$ fluxes were under the detection limit for all samples, i.e. below 11 g N-$N_2O$ ha$^{-1}$ d$^{-1}$. In this case, the reference $r_{N2O}$ values form the $^{15}$N treatment could not be precisely determined. However, from the information that $N_2$ flux is below the detection limit even for the highest $N_2O$ fluxes observed we can assess that $r_{N2O}$ must be higher than 0.75. For F1, soil temperature varied from 1.6 to 8.6 °C, mean 4.1 °C, WFPS varied from 54.1 to 72.4 %, mean 65 %.

The second field campaign F2 in March 2016 (7$^{th}$ March-11$^{th}$ March) showed very variable $N_2O$ fluxes from 0.5 to 110.7 g N-$N_2O$ ha$^{-1}$ d$^{-1}$. $N_2O$ isotopic signatures could be determined only in 17 samples from 26. The $N_2$ fluxes were above the detection limit for 15 samples from 26, and varied from 1 to 9 g N-$N_2O$ ha$^{-1}$ d$^{-1}$. In this case, the reference $r_{N2O}$ values form the $^{15}$N treatment could be determined for 4 sampling dates out of 8. For F2, soil temperature varied from 1.4 to 12.0 °C, mean 6.4 °C, WFPS varied from 57.9 to 77.9 %, mean 69 %.

The third field campaign F3 in Mai/June 2016 (30$^{th}$ Mai-3$^{rd}$ June) showed very high $N_2O$ fluxes from 1 to 1471 g N-$N_2O$ ha$^{-1}$ d$^{-1}$. $N_2O$ isotopic signatures could be determined in all samples. The $N_2$ fluxes were always above the detection limit and varied from 114 to 2060 g N-$N_2O$ ha$^{-1}$ d$^{-1}$. In this case, the reference $r_{N2O}$ values form the $^{15}$N treatment could be determined for all 8 sampling times. For F3, soil temperature varied from 17.0 to 32.5 °C, mean 21.4 °C, WFPS varied from 52.1 to 72.0 %, mean 62 %.

The detailed variations in gas fluxes during field campaigns and variations in $^{15}$N abundance in various pools ($a_{NO3}$, $a_{P\_N2O}$ and $a_{P\_N2}$) and the $N_2O$ $^{15}$N-pool derived fraction ($f_{P-N2O}$) are presented in the supplement (Fig. S2 C-E and Fig. S3 C-E). There are no significant differences in $N_2O$ flux between $^{15}$N and NA treatment (Fig. S2 C-E). In F3 the fluxes were much larger than in F1 and F2 and were decreasing during the sampling campaign, whereas $N_2$ flux was very variable and showed large differences between repetitions, represented by large error bars (Fig. S2 E). In F1 and F2 the $^{15}$N-pool derived fraction was significantly lower when compared to F3. In F3 $a_{P\_N2}$ and $a_{P\_N2O}$ was comparable and higher than $a_{NO3}$ in the first three samples and similar with $a_{NO3}$ for the last 5 samples. In F2 $a_{P\_N2O}$ strictly depended on $a_{NO3}$ and both showed clear decreasing trend, whereas $a_{P\_N2}$ was determined only in two sampling points and was significantly lower than $a_{P\_N2O}$ and $a_{NO3}$.




### 3.3 Laboratory experiments

The laboratory experiment L1 was conducted in dryer conditions than L2. In L1 initially WFPS was about 60 % and after water addition (9th day of the experiment) it was increased to 65%. In L2 initially WFPS was about 70 % and after water addition (9th day of the experiment) it was increased to 80 %.

$N_2O$ fluxes in L1 were quite low from 0.2 to 16.7 g N-$N_2O$ ha$^{-1}$ d$^{-1}$. $N_2O$ isotopic signatures could be determined in 38 from 56 samples. The $N_2$ fluxes were above the detection limit only for 43 from 112 samples and varied from 0 to 85 g N-$N_2O$ ha$^{-1}$ d$^{-1}$. In this case the reference $r_{N2O}$ values form the $^{15}$N treatment could only be determined for 7 sampling times out of 10. In L2 $N_2O$ fluxes were higher and varied in wide range from 0.4 to 297.4 gN-$N_2O$ ha$^{-1}$ d$^{-1}$. $N_2O$ isotopic signatures could be determined in 40 from 56 samples. The $N_2$ fluxes were

above the detection limit only for 87 from 112 samples and varied from 0 to 199 g N-$N_2O$ ha$^{-1}$ d$^{-1}$. In this case, the reference $r_{N2O}$ values form the $^{15}$N treatment could be determined for 9 sampling times out of 10.

The detailed variations in gas fluxes during laboratory incubations and variations in $^{15}$N abundance in various pools ($a_{NO3}$, $a_{P\_N2O}$ and $a_{P\_N2}$) and the $N_2O$ $^{15}$N-pool derived fraction ($f_{P-N2O}$) are presented in the supplement (Fig. S2 A-B and Fig. S3 A-B). We often observe significantly different fluxes for NA and $^{15}$N treatment: for L1

only for 2 samples (4 and 5) NA treatment show significantly higher $N_2O$ flux but for L2 majority of sampling points show significantly higher $N_2O$ flux in $^{15}$N treatment, particularly for the last 4 sampling points, after the water addition (Fig. S2 B). Importantly, water content did not differ for this sampling points. In L1 the $^{15}$N-pool derived fraction was significantly lower when compared to L2. In both L1 and L2 $a_{P\_N2}$, $a_{P\_N2O}$ and $a_{NO3}$ show comparable ranges and only very slight decreasing trend (Fig. S3 A-B).


Table 1 Results summary

### 3.5 Maps

For the graphical presentation of dual isotope plots for sampling points always $\delta^{18}$O and $\delta^{15}$N values of emitted

$N_2O$ are plotted ($\delta^{18}$O$_{N2O}$, $\delta^{15}$N$_{N2O}$). But the precursors isotopic signatures ($\delta^{18}$O$_{H2O}$, $\delta^{15}$N$_{NO3-}$, $\delta^{15}$N$_{NH4+}$) are taken into account by respective correction of mixing endmembers isotopic ranges (see Table S1). Hence, the precursor ranges represent the expected isotopic signatures of $N_2O$ originating from each pathway for the particular case study characterised by specific precursor isotopic signatures. Such approach allows for presenting all data in the common isotopic scales without presumption on the dominating pathway and dominating

precursor. In previous papers, where $\delta^{18}$O and $\delta^{15}$N related to precursors ($\delta^{18}$O$_{N2O/H2O}$, $\delta^{15}$N$_{N2O/NO3}$) were plotted (Ibraim et al., 2019; Lewicka-Szczebak et al., 2017; Lewicka-Szczebak et al., 2016) it was assumed that denitrification must be the dominating $N_2O$ production pathway.



**SP/O Map**

Fig. 1


The majority of isotope results presented in the SP/O Map (Fig.1) is situated within the area limited by reduction and mixing lines, which allows for application of the calculation approach based on SP/O Map. Numerous samples, mostly from the laboratory incubation studies, are situated below the mean reduction line but within the minimum reduction line. For these samples, the calculation results provide $f_{bD}$ values slightly above 1, which are

set for 1 for the further summaries. All calculations and results can be followed in the spreadsheet file in supplementary materials.

The endmembers isotope values applied here (after Yu et al. (2020)) differ for nitrification $\delta^{18}O$ when compared to previous applications of SP/O Map (Buchen et al., 2018; Ibraim et al., 2019; Lewicka-Szczebak et al., 2017; Verhoeven et al., 2019). The currently applied $\delta^{18}O$ endmember values for Ni (23.5 ± 2.1‰) are lower than

previously applied range (from 38.0 to 55.2 ‰, mean 43.0 ‰) and thus result in a separation of Ni and fD, which was not possible in the previous studies. With the current values, we have two possible mixing lines (bD-Ni and bD-fD), whereas in previous studies only one mixing line was applied (bD-(Ni+fD)). This requires the choice of most appropriate mixing scenario for the particular case study. For this study, the results obtained for $r_{N2O}$ and $f_{bD}$ differ mostly only very slightly for both mixing scenarios (see supplementary material, Table S2 and

spreadsheet file), which is due to high $f_{bD}$. For F3, where $f_{bD}$ is near 1, the difference in $r_{N2O}$ does not exceed 0.02, and for F1 with the lowest $f_{bD}$ of ca. 0.7, the difference in $r_{N2O}$ reaches 0.22 (Table S2). Below we summarize the results of calculations assuming bD-fD mixing scenario only.

The calculation has been performed with two cases (see Section 2.5) and all results are shown and compared with reference method in Table 2 and 3. Due to quite high $f_{bD}$ for our study the both cases show only very slight

differences (Table 2, Table3). For the field study F1 we obtained the highest $r_{N2O}$ values (0.86±0.12) and the lowest $f_{bD}$ values (0.74±0.07). For field study F2, the $r_{N2O}$ values were lower (0.38±0.05) and the $f_{bD}$ values were higher (0.92±0.04). For field study F3 the $r_{N2O}$ values were very similar as in F2 (0.33±0.07) and the highest $f_{bD}$ values were noted (0.99±0.01). For the laboratory incubation studies we obtained slightly lower ($p$=0.086) $r_{N2O}$ for L1 (0.19±0.03) when compared to L2 (0.27±0.12). Both laboratory treatment showed very high $f_{bD}$ for L1

(0.99±0.01) and L2 (0.98±0.04).

**3.6 SP/N Map**

Fig.2





For the SP/N Map we present the literature endmember values in relation to the respective precursor, i.e.
$NO_3^-$ for bD and fD and $NH_4^+$ for nD and Ni (supplement, Table S1). For the field and laboratory studies,
separate mean values for $NO_3^-$ (11.9 and 4.5 ‰ respectively) and $NH_4^+$ (41.4 and 79.3 ‰, respectively) were
applied. These precursor isotopic signatures are the means of 5 samplings for each campaign and experiment.
The extremely $^{15}N$ enriched $\delta^{15}N_{NH4}$ values result in large shift of endmember ranges for nD and Ni. These
ranges are $^{15}N$ depleted in relation to bD when assuming identical $\delta^{15}N$ values for $NO_3^-$ and $NH_4^+$, according to
most previous studies (Ibraim et al., 2019; Koba et al., 2009; Toyoda et al., 2011). But in the case of our
experiments, conversely, $N_2O$ originating from nD and Ni would be significantly enriched in $^{15}N$ when
compared to bD and fD (Fig. 2). For the samples the measured bulk $\delta^{15}N_{N2O}$ is plotted.

The majority of the samples is located outside the area limited by reduction and bD-fD mixing lines, which
mostly precludes the application of calculation approach based on SP/N Map. The separation of mixing and
reduction processes is not possible based on this plot, since the slopes of reduction line and bD-Ni mixing line
are too similar, especially for laboratory experiments (Fig. 2B).

Another approach to include N precursors values is to apply the individual endmembers isotopic signatures for
each $N_2O$ sample by interpolating the measured isotopic signatures of $NO_3^-$ and $NH_4^+$. With 5 measurements of
mineral N isotopic signatures per experiment we get quite a good resolution of these values. Since they show
quite high variations (Table 1) applying individual values is a better approach. But still, also by this approach the
majority of samples show values out of the calculation range and the results are very ambiguous representing the
whole range of possible variations in both $r_{N2O}$ and $f_{bD}$ values. Therefore these values are not summarized here.

**3.7 O/N Map**

Fig.3

For O/N Map (Fig.3) the $\delta^{18}O$ values for bD, fD and nD are expressed in relation to soil water and the $\delta^{15}N$
values for bD and fD in relation to soil $NO_3^-$ and for nD and Ni in relation to soil $NH_4^+$ (supplement, Table S1).
For these graphs, it is difficult to determine the reduction-mixing area because the slope of the reduction line is
almost identical to the bD-fD mixing line.

A significant linear correlations has been found both for the field and laboratory studies, with $R^2=0.27$ ($p<0.1$)
and $R^2=0.40$ ($p<0.01$), respectively. Both correlations show similar linear equations: $\delta^{18}O = 0.24* \delta^{15}N +33.3$
and $\delta^{18}O = 0.28* \delta^{15}N +41.6$, for field and laboratory studies, respectively (Fig. 3).

**3.8 3DI model**

The application of Maps applying $\delta^{15}N$ data, $i.e.$, SP/N and O/N Map, is very imprecise for this case study due to
untypically high $\delta^{15}N_{NH4}$ values and shifted location of the nD and Ni mixing endmembers (Fig. 2, Fig. 3).





However, still the $\delta^{15}N$ data comprise important information, which can assist in processes identification when applied jointly with the SP/O Map. Therefore, we combined all the information in one 3DI model where all three isotopic signatures are taken into account.

The results of this model regarding $r_{N2O}$ are mostly well comparable to the values obtained with SP/O Map
(Table 2). However, whereas for SP/O Map both Case 1 and Case 2 provide similar results for $r_{N2O}$, for 3DI model these differ more pronouncedly. On the pie diagrams (Fig. 4) the differences in the calculation assumptions for both cases can be visually compared. In Case 1, the $N_2$ fraction originates from $f_{bD}$ only, whereas in Case 2 it originates from all the fractions. Below we summarize the results of Case 2, which provides more reliable results, as further discussed (see Section 4.2).

We get much more detailed estimation regarding mixing proportions with 3DI model when compared to the SP/O Map. The dominating $N_2O$ production pathway is clearly bD, which contributes in $N_2O$ production from 46 % for F2 up to 69 % for L2 (Fig.4). An important role plays also nD contributing from 15% for L2 up to 40% $N_2O$ for F3; low $f_{nD}$ of 4% was found for F1. The $f_{fD}$ is quite variable from 6% for F3 to 26% for F1. Ni shows the lowest contribution around 3-5%, and only slightly higher $f_{Ni}$ of 13% was found for F2 (Fig.4). $N_2$ fluxes are
highly variable between the experiments, *i.e.,* mean $r_{N2O}$ values vary from 0.21 for L1 to 0.89 for F1 (Fig. 4, Table 2).

Fig. 4

The model provides very detailed information on probability distribution of the results, which is presented on the
matrix plots prepared after Parnell et al. (2013) (Fig. 5 shows example plots, all plots are shown in the supplement, Fig. S4), where histograms of probability distribution of $r_{N2O}$ and mixing proportions, correlations between the modeled fractions and R coefficients of these correlations are presented (Fig.5). This summary provides an overview of the reliability of the model outputs and allows for identifying unavoidable model inadequacy. For all the samples we observe very strong negative correlation between $f_{bD}$ and $f_{nD}$, similar for both
cases, from -0.28 to -0.93, mean -0.63, and between $f_{bD}$ and $f_{fD}$ from -0.15 to -0.97, mean -0.74. $r_{N2O}$ for Case 2 is always correlated negatively with $f_{bD}$ from -0.15 to -0.84, mean -0.62, and positively with $f_{fD}$ from 0.18 to 0.82, mean 0.62. For Case 1 this correlation is extremely variable for $r_{N2O}/f_{bD}$ from -0.67 to 0.85 and for $r_{N2O}/f_{fD}$ from -0.72 to 0.69. The lowest correlation coefficients are noted for $f_{Ni}$, where mean values never exceed 0.4. This is reflected in the determined ranges of possible results presented in the histograms. $f_{Ni}$ range is typically much
narrower than $f_{bD}$ and $f_{nD}$ ranges.

The correlations and histograms vary between the particular campaigns with some typical features. Therefore, in Fig. 5 we present a representative example of the correlation matrix plots for each campaign. The samples with complete repetitive measurements and lowest variations within the repetitions were chosen to present the most representative picture not affected by individual outliers. For F1 we observe a very similar output for Case 1 and





Case 2, quite narrow ranges of results and no extremely high correlations. For F2 the ranges are much larger and high negative correlations $f_{bD}$ / $f_{nD}$ and $f_{fD}$ / $f_{Ni}$ indicate possible imprecision in separation of these pathways, which results in much wider range of probable results. For F3 the most extreme negative correlation $f_{bD}$ / $f_{nD}$ is noted, and for Case 1 also r and $f_{nD}$ shows very strong correlation, which may affect the proper estimation of $r_{N2O}$. For L1 and L2 we observe lower correlation $f_{bD}$ / $f_{nD}$ but higher $f_{bD}$ / $f_{fD}$ which is probably a result of

different $\delta^{15}N$ endmember values for nD and Ni and better separation of these pathways. The strong positive correlation of $r_{N2O}$ and $f_{bD}$ for Case 1 in L1, F2 and F3 is rather a logical consequence of the assumptions underlying the Case 1 approach.

Fig. 5

### 3.9 Comparison of $r_{N2O}$ with independent estimates

The $N_2O$ reduction progress calculated with the above presented SP/O Map and 3DI model were compared with the results from the $^{15}N$ gas-flux method. In the tables below we present the detailed comparison with the results applying both calculation cases (Case 1 and Case 2) for $r_{N2O}$ (Table 2) and for mixing proportions (Table 3).

Table 2


The ranges and the mean values of the replicates means of all sampling dates are quite well comparable for SP/O Map and 3DI model Case 2. Most inconsistent results are obtained in Case 1 of 3DI model, however, for L2 this case seem to be most accurate.

Since the variations of $r_{N2O}$ values in the experiments are very variable in time just a comparison of overall mean

values is not informative, we need to compare the temporal changes of $r_{N2O}$ (Fig. 6).

Fig.6

Most extreme changes in time are reported for the laboratory experiment L2 where a very sudden change in $r_{N2O}$

was observed as a consequence of water addition (between sampling 5 and 6). All three estimates present the same trend as the reference method, however, with lower amplitude (Fig. 6B). For field study F3 $^{15}N$ treatment indicates a constant decrease in $r_{N2O}$, which is only partially reflected in SP/O Map and not at all in 3DI model results. F1 and F2 data are not complete due to $N_2$ fluxes under detection limit for the whole F1 sampling and half of the samples of F2 campaign. However, for this missing data we can make estimates of the $r_{N2O}$ based on

the known detection limit for $N_2$ flux. We estimated the $r_{N2O}$ values for the missing points assuming the possible $N_2$ flux: from 0 up to detection limit of 11.3 gN $N_2$ ha$^{-1}$ d$^{-1}$.





Fig.7

In Fig. 7 we checked the fit of $r_{N2O}$ values determined by [15]N gas-flux and 3DI model (Fig. 7A) or SP/O Map
        (Fig. 7B). When analysing all the individual sampling dates or all experiments, the fit to 1:1 line is not very well,
        especially for many dates of the L2 experiment $r_{N2O}$ is largely underestimated with isotopocule approaches. This
        is mostly due to the sudden change in $r_{N2O}$ as presented above (Fig. 6B). But when we compare the means of the
        whole experiment or the experimental phases before and after water addition for L1 and L2 (red points in Fig. 7),
the fit is much better with all points within the error of 0.15 for 3DI model. For SP/O Map the L2 mean after
        irrigation still shows larger disagreement.
        The agreement between isotopocule methods and reference method was statistically checked with $F$ value (Eq.
        17). The results for all means, minimal and maximal values are shown in Table 2. The statistically significant
        agreement was proved for SP/O Map ($p<0.1$) and Case 2 of 3DI model ($p<0.05$), whereas Case 1 of 3DI model
shows no agreement. Particular $F$ values calculated with all sampling dates means indicate no significant
        agreement ($F=0.13$ for F3, $F=0.45$ for L1, $F=0.28$ for L2 – values for fit between Case 2 of 3DI model and
        reference method), which reinforces the observation based on Fig.7, that only mean experimental values show
        good agreement with the reference method, but not the individual samplings.

### 3.10 Comparison of mixing proportions with independent estimates

The mixing proportions obtained by different approaches are much more complex to compare than $r_{N2O}$ due to
        the fact that each approach provides distinct information.
        - With the reference method – [15]N gas-flux – we determine the [15]N-pool derived fraction of $N_2O$ ($f_{P\_N2O}$),
          hence for the [15]$NO_3^-$ treatment this is the fraction of $N_2O$ originating from the labeled [15]$NO_3^-$ pool.
          Theoretically, this can be bD or fD. It was intended to use the [15]$NH_4^+$ treatment for the determination of
580          $N_2O$ fraction derived from $NH_4^+$ pool but due to rapid $NH_4^+$ turnover into $NO_3^-$, we deal with a highly
          [15]N-labeled $NO_3^-$ pool in the [15]$NH_4^+$ treatment and hence are not able to precisely separate these pools
          (results not shown).
        - With SP/O Map we determine the $f_{bD}$ fraction. But since in the SP/O Map bD and nD cannot be
          distinguished due to overlapping isotopic signatures (Fig. 1) this fraction actually informs about bD+nD
585          fraction.
        - With the 3DI model we are able to theoretically determine most of the fractions contributing to the $N_2O$
          flux, but the precision of such determination depends on the isotopic separation of particular pathways
          in 3D isotopocule plot. In our case study this separation is not very good, especially for $\delta^{15}N$ (see
          Section 3.6 and 3.7), hence this determination is associated with pronounced uncertainty (Fig.5).



To compare all this results we present a comparison $f_{P\_N2O}$ of $^{15}$N gas-flux (representing bD+fD) with $f_{bD}$ of SP/O
Map (representing bD+nD) and respective results ($f_{bD}, f_{bD+fD}, f_{bD+nD}$) of the 3DI model (Fig.8, Table 3).

Table 3

Fig. 8

The reasonable agreement in the ranges of values is obtained for experiments L1, L2 and F3, but a large
disagreement with the reference $^{15}$N gas-flux method is observed for field studies F1 and F2 (Table 3). For these
studies, extremely low $f_{P\_N2O}$ was found by the $^{15}$N gas-flux method, of 0.28 and 0.23, respectively. The time
dynamics are not very well reflected by various approaches (Fig.8). This is mostly visible in F3 (Fig. 8E) where
the $f_{bD}$ and $f_{bD+fD}$ show large variations between samplings from below 0.1 to above 0.9. These rapid changes
show much lower amplitudes according to the $^{15}$N gas-flux approach. The contribution of $f_{bD+nD}$ determined by
the 3DI model as well as $f_{bD}$ determined by the SP/O Map are much more stable in time, which is especially clear
for F3 (Fig. 8E), but also true for other campaigns (Fig.8).
For the mixing proportions the statistical agreement with $F$ value (Eq. 17) cannot be determined because the
fractions provided by various approaches do not precisely refer to the identical pathways contributions and are
not directly comparable.

## 4. Discussion

### 4.1 Mapping approaches for N$_2$O data interpretation – opportunities and limitations

So far the interpretations of N$_2$O isotope data are most commonly done with dual isotope plots. Whereas SP/N
and O/N plots were applied in numerous studies before (Kato et al., 2013; Koba et al., 2009; Opdyke et al., 2009;
Ostrom et al., 2007; Ostrom et al., 2010; Toyoda et al., 2011; Well et al., 2012; Yamagishi et al., 2007; Zou et
al., 2014) the usage of the SP/O plot is quite a new idea (Lewicka-Szczebak et al., 2017), but already used for
field studies (Buchen et al., 2018; Ibraim et al., 2019; Verhoeven et al., 2019). The recent work basing on
archival datasets with independent estimates of N$_2$ flux showed some weak accordance of the results of the SP/O
Map with independent estimates (Wu et al., 2019). However, the reasons are difficult to identify for archival
data. Here we present the performance of mapping approaches validated with independent estimates based on
$^{15}$N gas-flux method and try to identify potential problems.

The first challenge, especially for field studies, is obtaining complete datasets. This is due to limited sensitivity
of the isotopic measurements and a need for sufficient N$_2$O and N$_2$ flux. For our first field study (F1), N$_2$ flux
was under the detection limit and the $r_{N2O}$ values can thus not be fully compared. For the F2 field study we have





numerous missing data due to $N_2O$ or $N_2$ flux under detection limit, hence only a limited number of data can be compared. This may be the main reason (besides other discussed later – Section 4.4) for the weakest accordance of the results for F2. For this field study only four samples showed the $N_2$ flux above the detection limit and
these measured $N_2$ fluxes associated with the low $N_2O$ fluxes yield very low $r_{N2O}$ values. For samples with $N_2$ flux below the detection limit the estimated $r_{N2O}$ ranges show possibly also much higher values (Fig. 6D). Hence, possibly by missing the measurements of low $N_2$ fluxes we miss the higher $r_{N2O}$ values and our calculated means are not representative for the whole experiment (Table 2).

SP/O Map

The SP/O Map was proposed (Lewicka-Szczebak et al., 2017) after it was found that $\delta^{18}O$ of the $N_2O$ produced by bacterial and fungal denitrification is quite stable and together with SP may be useable for discrimination of these pathways (Lewicka-Szczebak et al., 2016; Rohe et al., 2014a). As O-precursor for bD, fD and nD the soil water is accepted, under the assumption of nearly complete O-exchange between water and denitrification
intermediates. The high extent of O-exchange during denitrification has been confirmed experimentally (Kool et al., 2009; Lewicka-Szczebak et al., 2016; Rohe et al., 2014b) and it results in a quite stable range for mixing endmember values for $\delta^{18}O$ for bacterial and fungal denitrification (Fig. 1). Importantly, due to higher isotope fractionation effect associated with subsequent reduction steps of $NO_3^-$ to $N_2O$ (i.e. removal of oxygen atoms, so called branching effect) during fungal denitrification, the ranges for $\delta^{18}O$ of bacterial and fungal $N_2O$ differ
significantly (Lewicka-Szczebak et al., 2016). Fungal denitrification shows very consequent high O-exchange and high fractionation during O-branching (Rohe et al., 2014b; Rohe et al., 2017), whereas bacterial denitrification is characterized in general by lower fractionation, but the differences in both fractionation and O-exchange between particular bacterial strains are large (Rohe et al., 2017). As a result of lower O-exchange showed by some bacterial strains, $\delta^{18}O_{NO3-}$ is also incorporated into produced $N_2O$. This complicates the
application of the proposed SP/O Map. It is not clear how large is the importance of such bacterial strains characterized by low O-exchange in soil communities. We assume it must be low, because soil incubation studies indicated so far mostly very high exchange rates (Kool et al., 2007; Kool et al., 2009; Lewicka-Szczebak et al., 2016). These studies covered in total 16 soils and only for two forest soils characterized by very low $N_2O$ emission the O-exchange was around 20 % (Kool et al., 2009), otherwise over 60 %, with mean of around 90 %
(Kool et al., 2009; Lewicka-Szczebak et al., 2016). Importantly, the range of $\delta^{18}O$ values determined for bacterial denitrification does not assume complete O-exchange but is determined for the soil samples of O-exchange varying in the range from 63 to 100% (Lewicka-Szczebak et al., 2016). Hence, based on current knowledge, this can be assumed typical for most soils and experimental conditions. Also in this study, quite a good agreement of the $r_{N2O}$ determined by the O/SP Map and the reference method (see Section 3.9) allows us to
confirm the general assumption underlying this calculation method.





SP/N Map

The application of dual isotope plot SP/N was initially proposed by Yamagishi et al. (2007) for ocean waters and by Koba et al. (2009) for groundwater studies. In open water bodies, the application of SP/N Map might be

effective due to relatively homogenous distribution of substrates in the sampled water volume and thus not biased by the spatial heterogeneity in $^{15}$N enrichment that can occur in soils due to the fractionation processes in soil microsites (Bergstermann et al., 2011; Cardenas et al., 2017; Castellano-Hinojosa et al., 2019; Lewicka-Szczebak et al., 2015; Well et al., 2012). The $\delta^{15}$N isotopic signatures of samples were corrected for $NO_3^-$ substrate only and for water studies this approach was well justified by the complete conversion of $NH_4^+$ to

$NO_3^-$ (Koba et al., 2009). This assumption was based on the low $NH_4^+$ concentration and should result in equal $\delta^{15}$N of $NH_4^+$ and $NO_3^-$, which allowed to put the whole data into a single $\delta^{15}N^{SP}$ - $\delta^{15}$N scheme. But for soil studies, due to multiple possible N substrates and difficulties to find a proper correcting strategy, later studies rather applied bulk measured $\delta^{15}$N without corrections (Kato et al., 2013; Toyoda et al., 2011). Up to now, the most appropriate approach of taking precursors into account is the recalculation of literature mixing endmember

values to the actually measured substrate values for each particular pathway, namely $NO_3^-$ for denitrification and $NH_4^+$ for nitrification (Zou et al., 2014). But this approach was not successful for this study (see Section 3.6). When endmember mixing areas where recalculated with the measured substrate isotope signatures, most of the sampling points were located outside the mixing-reduction area. This is most probably due to large variations in isotopic signatures of the substrates and the fact that the analyzed bulk $\delta^{15}$N values are not representative for the

actually utilized substrate pools due to spatial heterogeneity of fractionating processes as outlined above. Moreover, the range of values for $NH_4^+$ and $NO_3^-$ of our studies resulted in a very untypical location of endmember ranges for denitrification and nitrification on the Maps (Fig. 2, Fig. 3), hence the method is not really suitable for discriminating mixing of these pathways and $N_2O$ reduction for this particular study. This is due to the extremely high $\delta^{15}N_{NH4}$ values (even up to 100‰) which are associated with low $NH_4^+$ contents (Table

1). This indicates that the ammonium pool was highly fractionated and nearly exhausted.

O/N Map

After it was observed that $N_2O$ reduction results in the typical O/N slope of 2.6 (Menyailo and Hungate, 2006; Ostrom et al., 2007; Well and Flessa, 2009) the O/N Map was proposed for identification of significant $N_2O$

reduction based on the observed slope higher than 1 (Opdyke et al., 2009; Ostrom et al., 2007). However, it must be noted that in case of shifts in the isotopic composition of the N or O substrate the assessment of the importance of $N_2O$ reduction is not valid (Ostrom et al., 2010). This approach was well suited for short term controlled experiments, however for longer filed studies, where we deal with large variations of N substrates isotopic signatures, application of this approach appears problematic. We plotted our data in the O/N Map and





found a significant linear relationship for field and laboratory studies, both with a very similar equations. The observed slopes of 0.24 and 0.28, respectively, are much below 1 although the $N_2O$ reduction shows important contribution for these experiments (Table 2). Hence, this observed slope is rather due to change of active substrate pool or changes in the isotopic fractionation (Cardenas et al., 2017). This might be a result of changes in soil moisture during experiments (irrigation or rain episodes) and between the experiments and field

campaigns. The observed shift in $\delta^{15}N$ is ca. four times larger than for $\delta^{18}O$. We suppose that water addition intensified $N_2O$ production and this might have caused significant enrichment in active nitrate pool in soil microsites. For O isotopes intensified $N_2O$ production may result in slightly lower O-exchange, which may increase the $\delta^{18}O$ values as a result of incorporation of nitrate O signature (Lewicka-Szczebak et al., 2015; Rohe et al., 2017). Consequently, the isotope effects due to reduction are significantly interfered by shifts in $N_2O$

precursors dynamics. Since for this Map both N and O isotopes depend on the precursor isotopic signature and are significantly altered by the diffusion (Well and Flessa, 2008), the interpretations based on this Map are the most ambiguous.

### 4.2 Three-dimensional $N_2O$ isotopocule model – perspectives of this new approach

Such a model for interpretation of $N_2O$ isotopic data is proposed here for the first time. This model is based on

the Bayesian mixing models being well established and widely applied method in food-web studies to partition dietary proportions (Parnell et al., 2013; Phillips et al., 2014). But for $N_2O$ the determination of mixing proportion of different pathways contributing to $N_2O$ production is further complicated by $N_2O$ reduction which alters the final $N_2O$ isotopic signature. This additional parameter was incorporated into the model equations (eq. 10, 11). Moreover, it is still not clarified, if the reduction of $N_2O$ produced during bacterial denitrification only is

possible (Case 1) or also $N_2O$ from other pathways can be further reduced by bacterial denitrifiers (Case 2), hence both cases need to be considered. The model has a few advantages over the SP/O Map. First of all, it allows for including uncertainties of input data into the model and allows for assessment of the confidence intervals for the results. Moreover, theoretically the 3DI model allows for separation of four $N_2O$ production pathways, currently identified as the most relevant, within them $f_{fD}$, which is so far not distinguishable with other

isotopic methods (Wrage-Mönnig et al., 2018).

For our case studies, it has been shown that $\delta^{15}N$ values are not useful in dual isotope plots for quantitative estimations (Fig.2, Fig.3, Section 3.6 and 3.7) but are helpful to constrain mixing proportions when incorporated into the 3DI model. Since the model bases on probability distribution, it allows for providing estimates even for imprecise data, e.g. as in our case by difficulties in proper determination of $\delta^{15}N$ endmember ranges due to very

unstable precursor isotopic signatures.

The model outputs allow us to assess the quality of model performance and reliability of the results (Fig. 5, Section 3.8). From the uncertainty analysis provided by the model, we can determine the confidence intervals for



the estimated values (Fig. 6, Fig. 8). This is a total uncertainty resulting from all possible uncertainty sources due to: ranges of endmember values and fractionation factors, variations in $N_2O$ isotopic signatures for one sampling

date, and convergence of possible model results for three isotopic signatures. We are not able to separate these uncertainties in this study.

Another measure of model performance is given by the correlations between obtained results of all the modeled probable solutions (Fig. 5). Previous studies applying similar models interpreted the strong negative correlations between determined mixing proportions as inability of the model to distinguish these sources (Moore and

Semmens, 2008; Parnell et al., 2013; Phillips et al., 2014). We observe strong negative correlations between $f_{bD}$ and $f_{nD}$ for most cases. This may indicate the uncertainty in determination of these fractions due to the lack of isotopic separation of these processes in the $\delta^{15}N^{sp}/\delta^{18}O$ space (Fig. 1). But such a correlation is also expected if we deal with two strongly dominating sources, and the correlations between $f_{bD}$ and $f_{nD}$ are indeed highest for F3, where the fractions of other pathways are lowest. Nevertheless, for fractions showing high correlations,

presentation of the sum of these both pathways may be much more informative than separation between them. Therefore, we observe much more stable results for the sum of $f_{bD}$ and $f_{nD}$ than for $f_{bD}$ alone (Fig. 8). However, the large variations of $f_{bD}$ are not only the modeling artifact, since they reflect the variations noted with the reference method, which is especially clear for F3 (see Fig. 8E). In this case study, we can see that the variations of $f_{bD}$ are larger than in the reference method but similar dynamics of these variations can be observed.

With the model we can quantify the contribution of four pathways, however, there are so far no precise enough reference methods to validate these results (Wrage-Mönnig et al., 2018) (see Section 3.10). But are the provided estimates plausible? We can check with the most characteristic outcomes. For F1 the highest $f_{fD}$ values were noted (Fig. 4H). For this field study also the highest $r_{N2O}$ and the lowest $f_{bD}$ were noted with all the methods (Table 2, Table 3, Fig. 6C, Fig. 8C). Since for fD $N_2O$ is mostly the final product not further reduced to $N_2$

(Sutka et al., 2008), the higher $f_{fD}$ should result in higher $r_{N2O}$ values, which was noted for F1. The highest $f_{Ni}$ was noted for F2. In this field study, the soil ammonium content is clearly the highest and nitrate the lowest (Table 1), which indicates that nitrification can be more active here during the whole study campaign, when compared to the other experiments where we deal with large ammonium consumption at the very beginning of the experiments. This accordance of results allows us to suppose that the general trends in pathways mixing

proportions provided by the model is plausible.

### 4.3 Agreement in estimates of isotopocule approaches and independent estimates

In general, the both cases of SP/O Map and Case 2 of 3DI model show very similar results, whereas Case 1 of 3DI model indicates always higher $r_{N2O}$ values, hence underestimates $N_2$ flux (Table 2, Fig. 6). For the SP/O Map, the application of different calculation cases has little impact on the final results because both cases show

very high and quite stable $f_{bD}$. The contribution of bD is expressed jointly with nD for the SP/O Map, due to their





isotopic overlap (see Section 3.5). As a result the necessary assumption for the SP/O Map is the possible reduction of $N_2O$ originating from these both fractions bD and nD, also for Case 1. Conversely for 3DI model, these both fractions are separated and for Case 1 only bD fraction can be reduced. The $r_{bD}$ values obtained for Case 1 are very low (eg. 0.2 for F2 and 0.15 for F3) but when recalculated to $r_{N2O}$ (for comparison with other

results) they get high (eg. 0.58 for F2 and 0.54 for F3, Table 2) due to respective $f_{bD}$ values (see Eq. 12). Therefore, the $r_{N2O}$ determined by 3DI model Case 1 is very vulnerable to proper determination of $f_{bD}$. And this fraction is not very precisely determined, as we know from strong correlation found for $f_{bD}$ / $f_{nD}$ (see Section 4.2). Consequently, the imprecise separation of $f_{bD}$ and $f_{nD}$ is the reason for the biased $r_{N2O}$ values for Case 1 3DI model. This bias is not significant when we deal with very high $r_{N2O}$ fraction, as for F1 (Table 2) or for very high

and stable bD contribution, as for L2 (Table 2, Fig. 8B). For Case 2 the lack of precision in $f_{bD}$ and $f_{nD}$ determination do not largely affect $r_{N2O}$ results, since $N_2O$ originating from all pathways can be reduced in this case (Eq.11). Hence, in further discussion for 3DI model results we take into account Case 2 outputs only. This observation may also indicate that not only $N_2O$ from heterotrophic bacterial denitrification can be further reduced to $N_2$. Although previous studies suggested rather the Case 1 to be more accurate (Verhoeven et al.,

2019; Wu et al., 2019), our comparison indicates that Case 1 of the 3DI model underestimates the $N_2O$ reduction in most cases (Table 2). This may reinforce a recent discussion on nitrifier denitrification mechanisms assuming that heterotrophic bacterial denitrifiers are relevant in reducing $NO_2^-$ from nitrification (Hink et al., 2017). This would support the assumption that $N_2O$ from nD can be further reduced by bD pathway.

The largest discrepancy in $r_{N2O}$ between isotopocule approaches and reference method is noted for F2 (Table 2).

In this field campaign we deal with very low $N_2O$ fluxes and the reference method indicates very low $r_{N2O}$ values, i.e., very high $N_2O$ reduction rate. Moreover, for F2 the highest soil moisture of the field studies was noted (Table 1), which may result in inhibition of gaseous exchange. In these conditions, it is very probable that some of the produced $N_2O$ is completely reduced, and consequently, the isotopic information on its reduction is missed. Complete $N_2O$ reduction in soil microsites would result in overestimation of $r_{N2O}$ values by the $N_2O$

isotopocule approaches and this is what we observe in this case (Fig. 6D).

Pronounced discrepancies in mean values are also noted for L2 laboratory incubation (Table 2), which is due to rapid changes in $r_{N2O}$ resulting from water addition (Fig. 6B, Section 4.1). This rapid change is noted in both SP/O Map and 3DI model and in the reference method, but the $N_2O$ isotopocule results seem to react slower and with lower amplitude. $N_2O$ isotopocule approaches base on isotopic analyses of $N_2O$, whereas [15]N gas-flux

method base on the direct $N_2$ measurements. If $N_2O$ is partially stored in soil we may deal with delay in our observations or discrepancy in results. This indicates that individual sudden changes are not well monitored by the isotopocule approaches but the general mean values and changing trends are very well reflected (Table 2, Fig. 7).





Summary statistics for agreement between isotopocule approaches and reference method indicate significant fit

for SP/O Map, where both cases show very similar fit, and for 3DI model Case 2, where the best fit was

observed (Table 2). This agreement is much better than recently shown by Wu et al. (2019), where numerous

cases with very poor agreement between the results of O/SP Map and reference method have been found. That

study analyzed archival datasets, from which many experiments consisted of various experimental phases – like

anoxic and oxic or before and after fertilizer addition. This might have complicated the comparability of the

results. As shown by our study, the sudden changes in experimental conditions are differently reflected in the

results of both methods. Whereas the reference method based on direct measurements of $N_2$ flux reacts

immediately, results of isotopocule approaches show a certain delay, possibly due to accumulation of $N_2O$ in the

soil (Fig. 6B). But when we compare the mean values for each experimental phase, the agreement between both

methods is much better (Fig.7). Additionally, the former study included some experiments with glucose

amendment (Wu et al., 2019), which results in a very rapid N turnover and in consequence unstable pathways

contribution.

The source partitioning of $N_2O$ production seems much more problematic than of $r_{N2O}$ values. This is also more

difficult to be evaluated with the reference method since it yields only the sum of fD and bD, $i.e.$, it does not

distinguish these individual processes (see Section 3.10). We are also aware that the model may not be very

precise in separation of $f_{bD}$, $f_{nD}$ and $f_{fD}$, since they often show strong negative correlation (see Section 3.8 and

4.2). Taking these considerations into account, we can well understand the fractions contribution for L1, L2 and

F3, where the $f_{bD}$ fraction of SP/O Map and $f_{bD+nD}$ of 3DI model are comparable and $f_{bD+fD}$ of the 3DI model and

$f_{P\_N2O}$ of the $^{15}$N gas-flux method show similar range and trends (Fig. 8A, 8B, 8E). However, a large bias in

source partition is observed for F1 and F2 field studies. The $f_{P\_N2O}$ determined by $^{15}$N gas-flux method is much

lower than any fraction determined with isotopocule methods (Fig. 8C, 8D). The very low $f_{P\_N2O}$ fraction

indicates large contribution of $N_2O$ originating from unlabelled pool, since the $f_{P\_N2O}$ of the labeled $^{15}NH_4^+$

treatment was also comparably low (data not shown). This $N_2O$ may originate from organic N pool pathway

(Müller et al., 2014; Zhang et al., 2015) or chemodenitfication (Wei et al., 2019). These processes are not

included in the isotopocule methods hence cannot be accounted for. For these two field studies F1 and F2 we

deal with relatively low fluxes and low temperatures, thus the processes invisible for high flux situations may

play significant role here.

**4.4 Possible origins of inconsistency and potential improvements**

From the comparison of isotopocule approaches and the reference method we can identify the condition when

the calculation based on natural abundance $N_2O$ isotopes may be biased. The Maps applying $\delta^{15}$N value are very

vulnerable to changes in substrate isotopic signatures. When we observe large variations in soil $NO_3^-$, $NO_2^-$ or

$NH_4^+$ isotopic signatures such approach should rather not be applied.



Most problematic is the occurrence of $N_2O$ production pathways which are so far not investigated for their characteristic isotopic signature. This might be heterotrophic nitrification, co-denitrification or chemodenitrification, as supposed for our case studies F1 and F2. These less examined processes gain on

significance when the $N_2O$ fluxes are generally low, like in F1 and F2. Hence, for low $N_2O$ fluxes application of isotope Maps and 3DI model is less precise.

Recent literature suggest that the most vulnerable value for SP/O Map is the isotopic signature of the bD mixing endmember and this parameter should be best determined in focused experiments (Buchen et al., 2018; Wu et al., 2019). It was shown that a short-term anoxic experiment with $N_2O$ reduction inhibition with $C_2H_2$ favors bD

(Lewicka-Szczebak et al., 2017; Lewicka-Szczebak et al., 2016). Such an experiment could have been used for determination of isotopic signature of bacterial denitrification characteristic for the particular soil used in this study and narrow the range of mixing endmember for bD pathway. Unfortunately, when planning and conducting these studies we did not have this complete knowledge and missed to perform such parallel anoxic incubations, but this should be strongly recommended for further studies applying SP/O Map or 3DI model.

The determination of initial delta values ($\delta_0$), unchanged by $N_2O$ reduction might be also helpful in further constraining the isotope Maps. These $\delta_0$ can be obtained from the relation of $r_{N2O}$ determined by reference method and measured isotopic signatures (Lewicka-Szczebak et al., 2017). Unfortunately, this approach was not successful for our data, because no significant correlation between $r_{N2O}$ and isotopic signatures could be found. This indicates unstable endmembers mixing proportions or some problems with parallel experiments. This was

also the case in previous validation experimental study (Lewicka-Szczebak et al., 2017), where for oxic conditions the variations were too high to obtain significant correlation and determine the $\delta_0$ values. This shows that oxic experiments are not well suited for determination of isotopic signatures of particular mixing endmembers and should be always accompanied by more focused and stable anoxic incubations.

Further enhancement in performance of the isotope Maps could be attained if the experiments determining the

initial isotopic composition of mixing endmembers were performed with the soil collected parallel to particular experiments and the anoxic incubations were performed in the conditions similar to field conditions during the particular case study. Possibly from such experiments some subtle differences in characteristic endmember isotopic signatures would be detected. It can be supposed that such differences could be the reason for worse $r_{N2O}$ agreement with reference method for L2 and F2 (Table 2). It has been shown that the changes in initial $\delta^{18}O$

value of bacterial denitrification endmember has significant impact on the final results (Wu et al., 2019). We have checked if this could bring better agreement . For L2 the perfect agreement of SP/O Map and reference method is obtained when applying slightly higher $\delta^{18}O$ values: 25‰ instead of 19.3 ‰. Conversely for F2, much lower $\delta^{18}O$ values: 10‰ instead of 19.3‰ would be needed to obtain the perfect agreement. This differences are quite possible, the low values for F2 might be a result of low temperature and low fluxes, and in consequence

moderate or slow processes associated with maximal O-exchange. On the contrary, for high water content and





high temperature in L2 experiment we can expect slightly lower O-exchange resulting in higher initial $\delta^{18}O$ values.

## Conclusions

- It was shown that $N_2O$ residual fraction can be calculated based on isotope fractionation during $N_2O$ reduction with SP-$\delta^{18}O$ Mapping approach. The SP-$\delta^{15}N$ Mapping approach appeared more complex and problematic.

- Here we present for the first time the idea of applying triple isotope plot and develop a model based on all three $N_2O$ isotopic signatures. We are convinced that this is a powerful step forward in development of $N_2O$ isotopocule methods to quantify especially $r_{N2O}$, but also estimate some mixing proportions.

- Both $N_2O$ isotopocule based approaches - SP/O Map and 3DI model – show good accordance of $r_{N2O}$ with reference method and very comparable results to each other. For 3DI model the results of Case 2 (assuming possible $N_2O$ reduction of all $N_2O$ production pathways) were taken into account, since the results of Case 1 (assuming $N_2O$ reduction of bacterial denitrification only) underestimate the $N_2$ flux due to imprecision in determination of $f_{bD}$.

- The determination of mixing proportions with $N_2O$ isotopocule based approaches is biased for cases where additional processes not incorporated into the model occur. This may be the case when very low $N_2O$ fluxes are noted.

- $N_2$ flux determined from [15]N labelled treatments (reference method) show more rapid changes compared to values determined with $N_2O$ isotopocule approaches. Hence, the $r_{N2O}$ determined with $N_2O$ isotopocule approaches provides a good approximation of the averaged $N_2O$ reduction range, but do not reflect dynamic changes of $r_{N2O}$ with high resolution.

- For the 3DI model, the correlation matrix plots allow for a good control of the results quality, which is a clear advantage over the results provided with SP/O Map.

- According to these findings, the SP/O Map and 3DI model can be applied for $r_{N2O}$ determination with expected precision of around 0.1. For cases where the mixing proportions separation is imprecise, which can be supposed when model results show high negative correlations, the results should be carefully interpreted and preferably the values of correlated fractions should be shown jointly. In such cases, the calculation Case 2 should be applied for $r_{N2O}$ determination, since Case 1 incorporates possibly biased $f_{bD}$ into the final $r_{N2O}$ value. Importantly, even for these cases where the determination of mixing proportions was biased, we got reasonable estimates of $r_{N2O}$ values (with Case 2 calculations).



**Data availability.** Original data are available upon request. Material necessary for this study findings is presented in the paper and supplementary materials.


**Author contribution**. DLS and RW designed the field studies and laboratory experiments and DLS was in charge of caring them out. DLS performed the interpretations based on isotope mapping approaches and initiated the idea of three-dimensional model. MPL developed the model and provided results for analysed case studies with graphical presentations. DLS prepared the manuscript with significant contribution of RW and MPL.


*Competing interests.* The authors declare that they have no conflict of interest.

*Acknowledgements.* This study was financed by German Research Foundation (grant LE 3367/1-1 to DLS) and conducted in cooperation with the research unit 2337: "Denitrification in Agricultural Soils: Integrated Control
and Modeling at Various Scales (DASIM)" (German Research Foundation, grant WE 1904/10-1 to RW). Many thanks are due to Frank Hegewald and Nicolas Ruoss for help in conducting field studies, Stefan Burkart for help in carrying out soil incubation, Martina Heuer for help in isotopic analyses, Nicole Altwein and Ute Tambor for help in preparing laboratory incubation and in soil analyses, Kerstin Gilke for chromatographic analyses and Caroline Buchen for advice in preparing field campaigns.

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






**Tables and Figures**

**Table 1 Results summary**

|  | treatment | F1 | F2 | F3 | L1 | L2 |
|---|---|---|---|---|---|---|
| WFPS [%] |  | 65.1 ±4.3 | 69.1±4.5 | 62.4±4.1 | 60→65 | 70→80 |
| $N_2O$ flux | NA | 8.9±7.4 | 16.3±26.1 | 331.3±302.9 | 4.9±4.7 | 8.5±5.6 |
| [gN-$N_2O$ ha$^{-1}$d$^{-1}$] | $^{15}$N | 5.9±5.5 | 4.3±3.3 | 330.9±323.7 | 1.4±1.0 | 54.6±50.2 |
| $N_2$ flux[a] | $^{15}$N | bd (<11.3) | 108.2±84.1[b] | 576.4±285.4 | 23.3±19.2 | 43.4±44.5 |
| [gN-$N_2$ ha$^{-1}$d$^{-1}$] |  |  |  |  |  |  |
| $r_{N2O}$[a] | $^{15}$N | nd (>0.75) | 0.06±0.04[b] | 0.33±0.15 | 0.12±0.10 | 0.49±0.31 |
|  |  |  |  |  |  |  |
| $NO_3$ content | NA | 13.6±3.1 | 8.0±2.4 | 13.6±3.2 | 21.2±1.5 | 21.0±1.7 |
| [mg N kg$^{-1}$ soil] | $^{15}$N | 15.8±6.2 | 7.5±1.1 | 15.8±5.5 | 20.1±0.6 | 19.4±1.1 |
| $NH_4$ content | NA | 3.8±2.1 | 6.4±3.3 | 3.4±1.5 | 0.53±0.19 | 0.71±0.23 |
| [mg N kg$^{-1}$ soil] | $^{15}$N | 2.0±2.6 | 5.4±3.1 | 3.7±1.9 | 0.58±0.2 | 0.72±0.15 |
| $\delta^{15}N_{NO3}$ [‰] | NA | 8.0±5.4 | 11.7±5.3 | 12.1±3.7 | 4.5±0.4 | 4.7±0.55 |
| $\delta^{15}N_{NH4}$ [‰] | NA | 31.0 ±8.7 | 40.5±6.8 | 42.2±9.1 | 90.0±7.9 | 70.4±17.9 |
| $a^{15}N_{NO3}$ [atom %] | $^{15}$N | 20.5 ±9.6 | 40.3±10.1 | 19.7±5.8 | 13.6±0.7 | 13.9±0.8 |
| $a^{15}N_{NH4}$ [atom %] | $^{15}$N | 0.7 ±0.6 | 0.9±0.4 | 0.5±0.2 | 0.5±0.03 | 0.5±0.01 |
| $a^{15}N_{NO2}$ [atom %] | $^{15}$N | 15.5 ±9.4 | 21.9±8.0 | 10.9±2.3 | 8.5±6.1 | 10.3±3.8 |
| $\delta^{15}N_{N2O}$ | NA | -33.4 ±9.5 | -20.2±16.0 | -14.0±14.8 | -2.4±8.0 | -17.7±11.9 |
| $\delta^{18}O_{N2O}$ | NA | 22.7 ±4.3 | 33.2±5.6 | 33.4±6.1 | 40.8±5.5 | 36.8±5.2 |
| $\delta^{15}N^{SP}_{N2O}$ | NA | 9.4 ±4.5 | 11.6±5.4 | 6.9±5.2 | 9.0±6.2 | 8.6±3.1 |
| $a^{15}N_{N2O}$ [atom %] | $^{15}$N | 7.5 ±2.7 | 11.7±7.3 | 16.2±10.6 | 11.8±0.72 | 13.7±0.67 |
| $f_{P\ N2O}$ | $^{15}$N | 0.28 ±0.12 | 0.23±0.13 | 0.59±0.19 | 0.69±0.06 | 0.96±0.09 |
| $a_{P\ N2O}$ | $^{15}$N | 0.28 ±0.07 | 0.47±0.09 | 0.26±0.11 | 0.17±0.02 | 0.15±0.01 |
| $a_{P\ N2}$ | $^{15}$N | nd | 0.23±0.11 | 0.33±0.11 | 0.21±0.07 | 0.18±0.06 |

[a] determined in $^{15}$N treatments with gas-flux method

[b] half of data below detection limit

bd – below detection limit

nd – not determined – due to $N_2$ flux below detection limit






**Table 2: Comparison of N₂O residual fraction ($r_{N2O}$) determined with the N₂O isotopocule approaches (SP/O Map and 3DI model) and the reference method ($^{15}$N gas-flux). Minimal (min), maximal (max) and mean values were calculated with the each sampling mean values (of all replicates). The agreement with the reference method was assessed with the Nash–Sutcliffe efficiency ($F$, Eq. 17) (Nash and Sutcliffe, 1970), which represent the $R^2$ of the fit to the 1:1 line (Fig. 7).**


| | | N₂O isotopocule approaches | | | | reference method |
|---|---|---|---|---|---|---|
| | | SP/O Map | | 3DI model | | $^{15}$N gas-flux |
| | | Case1 | Case2 | Case1 | Case2 | |
| L1 | min | 0.15 | 0.14 | 0.41 | 0.16 | 0.03 |
| | max | 0.24 | 0.24 | 0.71 | 0.32 | 0.30 |
| | mean | **0.19** | **0.18** | **0.49** | **0.21** | **0.12** |
| L2 | min | 0.16 | 0.15 | 0.40 | 0.17 | 0.12 |
| | max | 0.52 | 0.53 | 0.71 | 0.68 | 0.93 |
| | mean | **0.27** | **0.27** | **0.49** | **0.36** | **0.50** |
| F1 | min | 0.68 | 0.70 | 0.89 | 0.87 | 0.75[a] |
| | max | 1.00 | 1.00 | 0.93 | 0.93 | 1[a] |
| | mean | **0.86** | **0.86** | **0.91** | **0.89** | **nd[a]** |
| F2 | min | 0.30 | 0.36 | 0.46 | 0.22 | 0.02[b] |
| | max | 0.43 | 0.49 | 0.72 | 0.61 | 0.11[b] |
| | mean | **0.38** | **0.42** | **0.58** | **0.39** | **0.06[b]** |
| F3 | min | 0.26 | 0.27 | 0.39 | 0.27 | 0.17 |
| | max | 0.47 | 0.47 | 0.82 | 0.42 | 0.59 |
| | mean | **0.33** | **0.32** | **0.54** | **0.34** | **0.33** |
| agreement with reference method ($F$) | | **0.59\*** $p$=0.091 | **0.61\*** $p$=0.081 | **-0.09** | **0.77\*\*** $p$=0.015 | |

[a] all N₂ fluxes under detection limit, the range of values estimated based on detection limit – values not included in the statistics

[b] data not complete due to half of N₂ fluxes under detection limit – values not included in the statistics






**Table 3 Comparison of N₂O fraction originating from bD ($f_{bD}$) determined with the N₂O isotopocule approaches (SP/O Map and 3DI model) and the reference method ($^{15}$N gas-flux). Due to methodical assumptions for the particular approach either bD+nD fraction (for SP/O map and 3DI model) or bD+fD fraction (for 3DI model and reference method) can be compared (see Section 3.10).**

| | | N₂O isotopocule approaches | | | | | | reference method |
|---|---|---|---|---|---|---|---|---|
| | | SP/O Map (bD+nD) | | 3DI model (bD+nD) | | 3DI model (bD+fD) | | $^{15}$N gas-flux (bD+fD) |
| | | Case1 | Case2 | Case1 | Case2 | Case1 | Case2 | |
| L1 | min | 0.96 | 0.79 | 0.86 | 0.84 | 0.35 | 0.34 | 0.64 |
| | max | 1 | 1 | 0.94 | 0.94 | 0.71 | 0.71 | 0.75 |
| | mean | **0.99** | **0.93** | **0.89** | **0.89** | **0.59** | **0.59** | **0.70** |
| L2 | min | 0.94 | 0.88 | 0.65 | 0.66 | 0.65 | 0.65 | 0.81 |
| | max | 1 | 1 | 0.95 | 0.95 | 0.97 | 0.97 | 1 |
| | mean | **0.98** | **0.96** | **0.84** | **0.84** | **0.82** | **0.82** | **0.95** |
| F1 | min | 0.62 | 0.55 | 0.52 | 0.52 | 0.85 | 0.85 | 0.08 |
| | max | 0.84 | 0.83 | 0.82 | 0.82 | 0.97 | 0.97 | 0.42 |
| | mean | **0.74** | **0.70** | **0.70** | **0.70** | **0.91** | **0.91** | **0.28** |
| F2 | min | 0.84 | 0.64 | 0.62 | 0.59 | 0.34 | 0.14 | 0.16 |
| | max | 0.95 | 0.89 | 0.83 | 0.83 | 0.94 | 0.95 | 0.31 |
| | mean | **0.92** | **0.77** | **0.75** | **0.74** | **0.65** | **0.59** | **0.23** |
| F3 | min | 0.97 | 0.92 | 0.87 | 0.86 | 0.21 | 0.06 | 0.41 |
| | max | 1 | 1 | 0.93 | 0.93 | 0.92 | 0.92 | 0.83 |
| | mean | **0.99** | **0.97** | **0.90** | **0.90** | **0.60** | **0.56** | **0.59** |


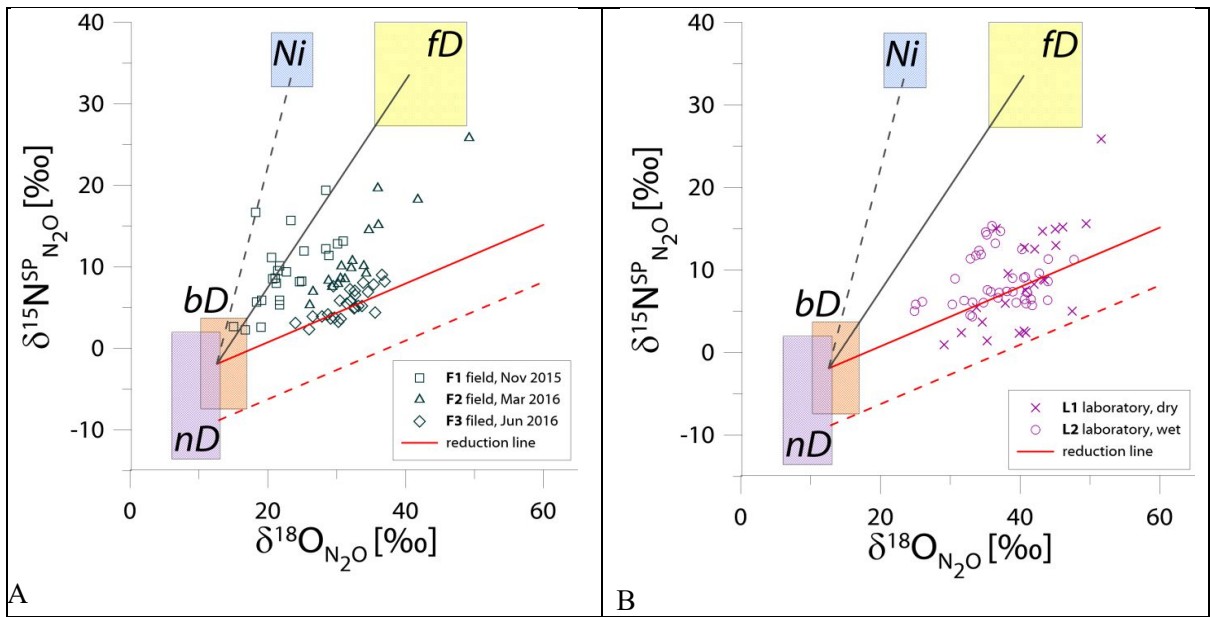

**Fig.1 N₂O isotope data of field (A, green points) and laboratory studies (B, purple points) in SP/O Map presented with literature endmember values and theoretical mixing (grey line) and reduction (red line) lines. $\delta^{18}$O values of mixing endmembers bD, nD and fD are presented in relation to the mean measured ambient water of -6.4‰ (hence present the expected $\delta^{18}$O$_{N2O}$ originating from particular pathway in this study conditions).**


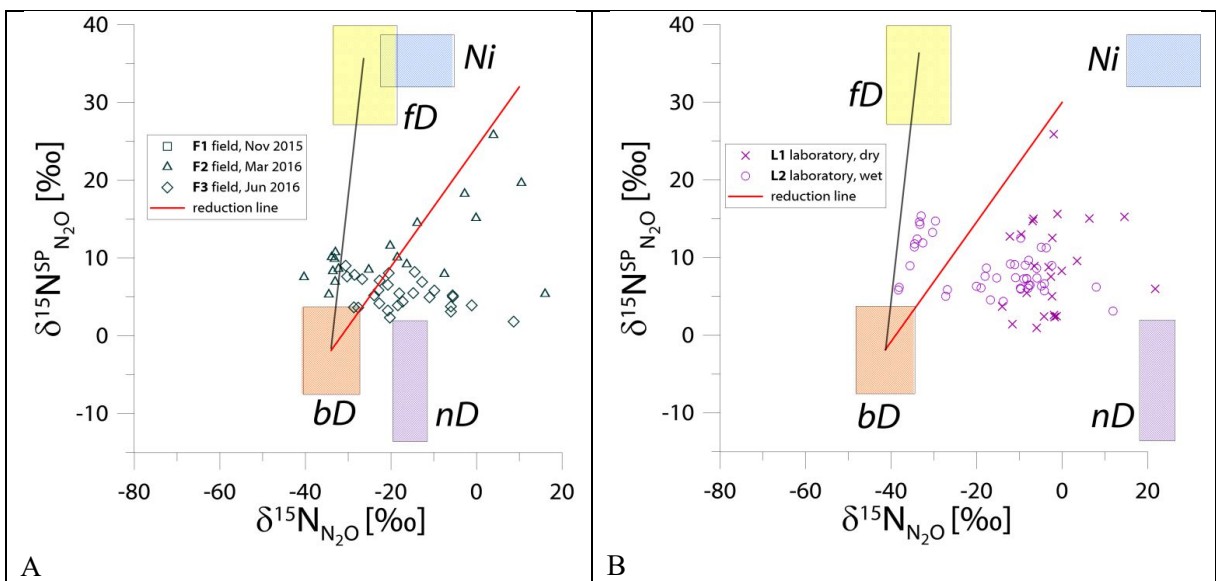

**Fig. 2 N₂O isotope data of field (green points) and laboratory (purple points) in SP/N Map presented with literature mixing endmember values and theoretical mixing (grey line) and reduction (red line) line. $\delta^{15}$N values of mixing endmembers are presented in relation to the $\delta^{15}$N of precursors: soil nitrate for bD and fD or ammonium for nD and Ni (hence present the expected $\delta^{15}$N$_{N2O}$ originating from particular pathway in this study conditions).**






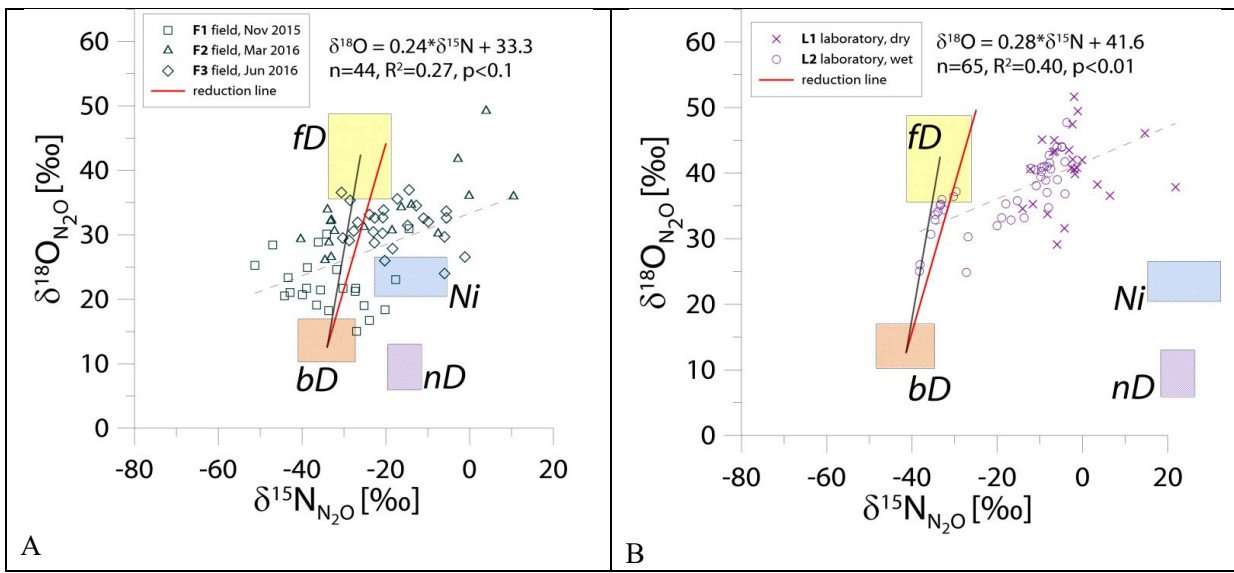

**Fig. 3** $N_2O$ isotope data of field (A, green points) and laboratory (B, purple points) in O/N Map presented with literature mixing endmember values and theoretical mixing (grey line) and reduction (red line) lines. $\delta^{15}N$ values are presented in relation to the $\delta^{15}N$ of precursors: soil nitrate for bD and fD or ammonium for nD and Ni. $\delta^{18}O$ values of mixing endmembers bD, nD and fD are presented in relation to the mean measured ambient water of -6.4‰. Hence, the mixing endmember ranges present the expected $\delta^{15}N_{N2O}$ and $\delta^{18}O_{N2O}$ originating from particular pathway in this study conditions.







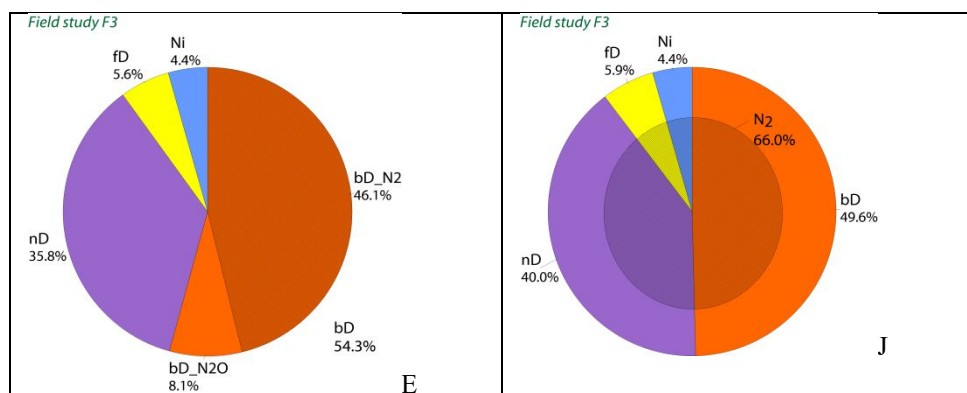

**Fig. 4** Pie diagrams of modeled mixing ratios ($f_{bD}$, $f_{nD}$, $f_{fD}$, $f_{Ni}$) and N$_2$ flux contribution in the total (N$_2$+N$_2$O) flux (1-
$r_{N2O}$). Results for both modeling cases: Case 1 (A-E) and Case 2 (F-J) are shown. Different graphical presentation of
N$_2$ contribution reflects the different assumption for both cases: N$_2$ can be produced only from bD in Case 1, but N$_2$O
from all pathways can be reduced to N$_2$ in Case 2. For both cases the percentage of N$_2$ is expressed in relation to the
total (N$_2$+N$_2$O) flux.


| F1 Case1 | F1 Case2 |
|---|---|
| F2 Case1 | F2 Case2 |
| F3 Case1 | F3 Case2 |





**Fig. 5 Matrix plots presenting detailed 3DI model outputs for each sampling date – here representative examples for each sampling campaign are shown (in the supplement plots for all samples are shown. Fig. S4). The plots in the diagonal show histograms of posterior probability distribution of $r_{N2O}$ and mixing ratios (scale from 0, left to 1, right), the plots above the diagonal show correlations between the modeled fractions (scale from 0, left to 1, right) and the values below the diagonal show $R$ coefficient of these correlations: in blue for positive correlations and in red for negative correlations with the size proportional to the $R$ value.**








**Fig.6 Comparison of time changes in residual N$_2$O fraction ($r_{N2O}$) determined with O/SP Map Case 1 and 3DI model with the reference method ($^{15}$N gas-flux). For the 3DI model results the 95% confidence interval is shown with grey shaded areas. Error bars for O/SP Map and $^{15}$N gas-flux data represent the standard deviation of replicate samples (n=4). For N$_2$ fluxes below the detection limit the estimated $r_{N2O}$ values are shown (red areas), calculated with N$_2$ flux**
**from 0 to 1 of the detection limit.**





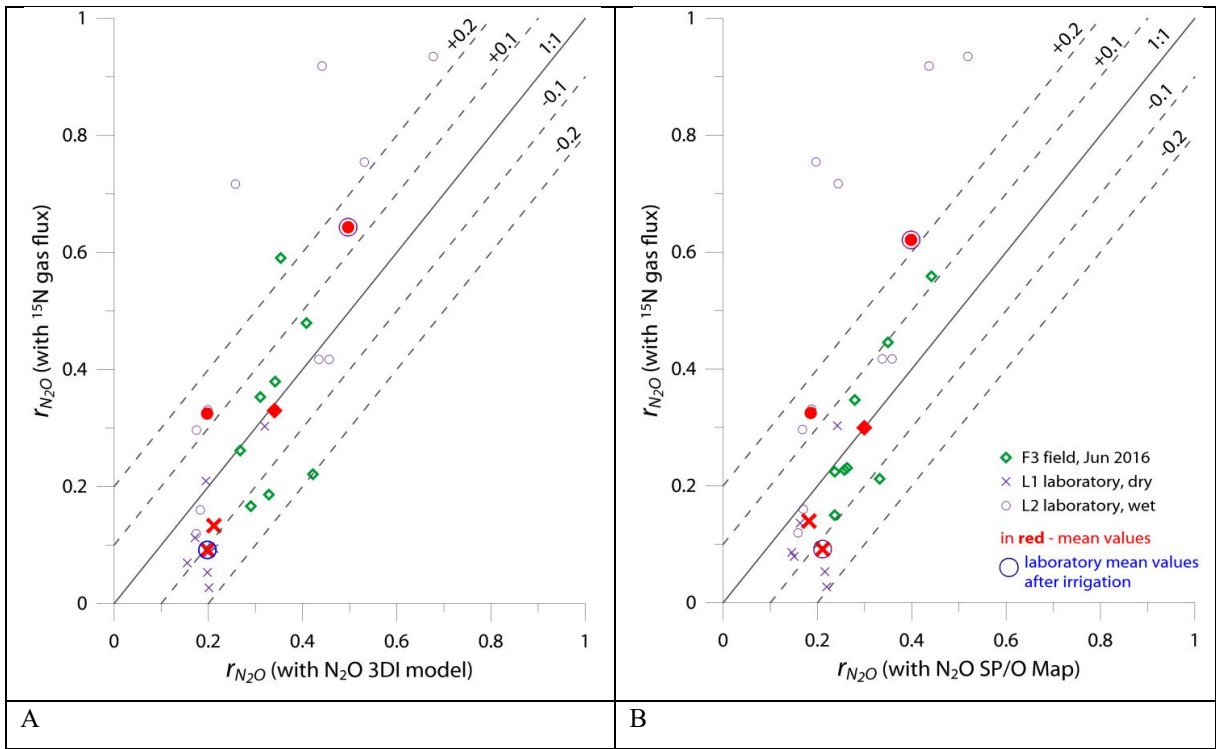

**Fig.7 Comparison of 1:1 fit between $r_{N2O}$ determined with the reference method ($^{15}$N gas-flux) and (A) 3DI model Case 2, (B) SP/O Map Case 1.**





**Fig.8 Comparison of N$_2$O fractions comprising bacterial denitrification ($f_{bD}$) determined with O/SP Map Case 1 (representing bD+nD) and 3DI model Case 2 (respective fractions determined: bD, bD+nD, bD+fD) with the reference method ($^{15}$N gas-flux).** $^{15}$N gas-flux method determines the $f_{P\_N2O}$ – $^{15}$N-pool derived fraction – comprising all N$_2$O origins utilizing $^{15}$N-labelled NO$_3^-$ – theoretically mostly bD and fD. See Sections 4.2 and 4.3 for further discussion. For the 3DI model results the 95% confidence interval is shown with shaded areas. Error bars for O/SP Map and $^{15}$N gas-flux data represent the standard deviation of replicate samples (n=4).

