# Peer review of "$N_2O$ isotope approaches for source partitioning of $N_2O$ production and estimation of N2O reduction – validation with 15N gas-flux method in laboratory and field studies"

_Biogeosciences, 2020_

## Referee Comment (RC1) · Anonymous Referee #1 · 3 Aug 2020

Review Lewicka-Szczebak et al. "N2O isotope approaches for source partitioning of N2O production and estimation of N2O reduction- validation with 15N gas-flux method in laboratory and field studies"

The study by Lewicka-Szczebak et al. compares different approaches for N2O source partitioning based on isotopomer measurements and also compares the so-called mapping approaches with experimental data. The paper is well written. The authors have so far coined the tool box for interpretation of N2O isotopomer data. With this study, they advance this branch of research by presenting a method that allows to

incorporate SP, 18O and 15N values into one sound approach. The paper is definitely an appreciated and valuable contribution to the scientific community and should be made accessible through a journal with wide audience like BG. I have only minor comments. One more relevant is summarized here (and specified in the detailed section), the others can be found in the detailed section below.

In the results section, the authors present an approach of correcting endmember isotope values obtained from the literature for the specific conditions of the given experiments or sites. As far as I understand the correction, rN2O values that are calculated based on 18O/SP maps are invariant towards this correction. The authors don't get back to the value of this correction compared to showing DELTAdelta maps or using DELTAdelta values and the reported isotope endmembers in the literature.

See some more detailed comments below. Title good Abstract good Introduction L29: please change to "important for"

L32: Determination instead of Partition?

L37: partition sounds odd to me at this place. I suggest referring to it as "determination". You don't do the partition. You want to determine it?

L45: pathways instead of pathway

L70: I am not sure if I understand correctly. The observed isotope effect for N2O reduction is quite stable and doesn't depend on r N2O. Do you mean "change in N2O isotopic composition due to N2O reduction depends largely on y n2o"?

L77-78: I don't understand "N2O mixing ratio between two N2O production pathways and rN2O". Section 2.1: has this experiment been conducted in 2015/2016. Please confirm if this is the procedure that has been applied in the past and state clearly that this paper reports on past campaigns and current incubations of the same soil.

Materials and Methods

L192 cf: On the one hand, water was added to obtain a certain water content in the

soil. On the other hand, columns were continuously flushed with He/O2/N2. These gases are dry (i.e., very low dew points), so that they will take up significant amounts of water from the soil, considering a flush time of 5 days. As a consequence, the soil columns will dry out under this treatment. How did the authors account for this drying effect when referring to the WFPS values? Are the WFPS values an average or the initially values that was aimed at?

L210: I suggest mentioning that fluxes were calculated based on the dynamic chamber principle. Correction for the inlet concentration is omitted since the gas mixture used for flushing is N2O-free. Do I get this right?

L310: Please give a rationale for expecting higher fungal denitrification than Ni

L319: I sugges explaining what the cases are: Case 1 assumes reduction of N2O produced through bacterial denitrification and subsequent mixing, whereas case 2 assumes mixing and subsequent reduction of the mixed pool by denitrifiers. . . . if I am getting you right.

L320: please explain c, and explain what happens if the condition is met.

Results

L421-425: Actually, I can't follow this short statement entirely. I understand that the authors aim at getting rid of the implicit assumption of DELTAdelta maps that the dominant source of N2O is denitrification. I thank the authors for this advancement. Some unclarities remain however:

1. For a 18O/SP map, the endmember values for bD, nD and fD are changed by the same amount, -6.4 per mil of water. Correcting for 18O of water makes sense, if nitrate exchanges O with water completely. However, when looking in the literature cited, it seems like the endmember values in the literature have been corrected for 18O already. For this reason, and I want to discuss this with the authors, I suggest correcting for the difference in d18O water between the presented study and the body of literature. Please clarify.

2. Again, for a 18O/SP map, this correction is equivalent with a correction of the

measured N2O 18O signature with water 18O, as correcting all endmembers for the same value is mathematically the same as correcting the measured values.

3. For 15N/SP maps, this is actually a further development, since this allows correcting both Ni and bD, fD and nD endmembers, in contrast to only correcting measured values with one assumed precursor composition.

I am asking the authors to comment on this and make this clear in the text if they agree. Please also explain the sign of your correction. It interferes with the definition of the apparent isotope effect in your reference 4 of the supplementary material (Sutka 2006).

L434: Please define minimum reduction line. I guess it is the dashed red line, but it hasn't been defined in the figure caption.

L461: there is a large discrepancy of field and laboratory NH4 delta values. This observation should also be taken up in the discussion again.

L495: In my opinion, the pie charts are i) not well described and ii) consume more space than necessary. If the authors stick to the pie charts, I ask them to indicate that the share of N2 produced during N2O reduction is given as N2 in percent. Hatched superimposed patterns show the source processes for N2O that has been reduced to N2. All in all, my suggestion is showing 2 bar plots (case 1 and case 2) with categories bD (bdN2O, bdN2), nD, fD, Ni, N2.

Figure 5: figure 5 gives everything but an overview of the results. I suggest showing only one case for Laboratory or field and instead have the numbers intelligible, and annotate scales. Please only show significant correlations, and don't correlate font size with correlation strength.

L569: why is p<0.1 considered as significant? I suggest rephrasing the sentence, proved sounds odd in terms of statistical inference. The p agreement between reference method and SP/O Map (p<0.1??) as well as Case 2 of 3DI model (p<0.05) was statistically significant.

Discussion

L718-720: could the authors please explain how the uncertainty was actually transferred in the model. In line 331-332, the authors state that "...uncertainties of the source's data ifs fed into the model through the variance in the calculation of unnormalized likelihood". However, I don't understand then the following sentence, that the prior distribution was assumed uninformative. Does this mean that for each endmember isotopic composition, the whole range of values was allowed? Please clarify how this actually works.

Conclusion
L863-864: I think this needs to be rephrased. The authors haven't presented a triple isotope plot, but rather present a likelihood function that allows for including three isotopic quantities and 4 associated isotopic endmembers.
* * *

---

## Referee Comment (RC2) · Anonymous Referee #2 · 1 Sep 2020

The manuscript by Lewicka-Szczebak et al. touches on an important topic developing within the efforts to more comprehensively understand the use of bulk and site specific N2O isotopes in delineating the extent of N2O reduction in order to more accurately apportion N2O production pathways. More importantly, the natural abundance N2O isotopes method is cross-checked with an independent approach (i.e. 15N-enriched experiment) - a combined study which is still generally lacking. The authors are commended for their meticulous and systematic considerations of the approaches used in their study. The extensive discussion on the suitability of the different approaches

(i.e. mapping versus model incorporating all three SP, d18O and d15N values) in both lab and field studies will definitely provide useful guidelines for similar studies in the future. As such, this study is topical, relevant and certainly fits the remit of BG. Generally the manuscript is well written and well-presented except in some places, more explanations/justifications are required (which I detail below):

Line 61: It would be helpful to provide the actual value of sensitivity increase here so that a direct comparison can be made.

Line 68: 'budget' is a more appropriate word here rather than emission. It is probably much easier to measure N2O fluxes directly if emission is intended.

Line 76: Should diffusion of N2O be taken into account as one of the processes determining the final N2O isotopes? Either here or in discussion, why the fractionation factor of diffusion is not considered in this study should be briefly mentioned.

Line 98: What was the non-identical treatment here? Suggest briefly describe to give the readers some idea on the treatment differences which should be avoided

Line 117: Suggest the authors indicate their reference method here

Line 159: Should also briefly mention what is the same treatment strategy employed in this study.

Line 178: Why 20 mg N/kg of soil in lab incubation compared to 10 mg N/kg soil in field fertilisation?

Line 141: This is confusing. Why the dates for 'next field campaigns' do not correspond to what is written earlier?

Line 217: How was N2O converted to N2? In-line conversion?

Line 298: nD is not included here. Why? Understand that the isotopic ranges are not very different between bD and nD but authors should briefly mention why this is not included here to avoid confusion. Also to show that the authors have considered the

nD pathway as well.

Line 310: Why higher fungal DN compared to nitrification in this instance? The data in Table S2 can be used to briefly justify this?

Line 379: Some of the N2 fluxes are above the detection limit but the values written here are below the detection limit mentioned in line 373.

Line 421 - 423: Don't quite get what you mean here. Consider rephrasing. You mean corrected precursor ranges based on different fractionation factors?

Line 434: The minimum reduction line is not described in Fig. 1. The dotted lines and the mixing should be clearly described in the legend/caption.

Line 461: There was relatively large discrepancy between lab and field NO3- and NH4+ values. In fact, the d15N-NH4+ is very heavy and the possible factors driving these values should be discussed.

Line 490: The authors mentioned that the high d15N-NH4+ has shifted the location of the nD and Ni in the end member mixing plot. What is the author comparing the shift to?

Line 551: Amplitude for 3D1 model, case 1 is not always lower than the reference – at the start and towards the end of sampling, the amplitude is higher than the reference method. Any explanation on why this is the case?

Line 670: I agree with the authors that recalculation of the literature mixing endmember values is important but my question is what fractionation factors should be considered when correcting these values and how to evaluate that these corrected ranges are justified?

Line 686: Be specific of what shift is meant here? Temporal?

Line 820: This sentence is rather subjective. Is it possible to provide a more definitive range here? Can the authors make use of a sensitivity analysis to show the extent of

substrate isotopic variations effects on the accuracy of the mapping approach?

Line 825: Can the author suggest the lowest N2O fluxes without compromising the precision of isotope maps and the 2DI model? This will be helpful as a guideline for future studies wanting to use these approaches.

———————————————————

---

## Author Comment (AC1) · 18 Sep 2020

*Author's response on*

**"N₂O isotope approaches for source partitioning of N₂O production and estimation of N₂O reduction – validation with ₁₅N gas-flux method in laboratory and field studies" by Dominika Lewicka-Szczebak et al.**

**Review response for Anonymous referee #1**

- (1) *comments from referees*
- (2) authors response
- (3) authors changes in manuscript

Thank you very much for your positive evaluation of the manuscript and your critical comments which helped us to improve our work.

We will take into account all the comments and especially clarify the endmembers isotope values corrections we applied. Here we provide detailed responses for the raised points, minor corrections have been accepted and will be included into reviewed manuscript.

*L70 I am not sure if I understand correctly. The observed isotope effect for N2O reduction is quite stable and doesn't depend on r N2O. Do you mean "change in N2O isotopic composition due to N2O reduction depends largely on y n2o"?*

Isotope fractionation during N2O reduction is independent of rN2O (mostly). But with the isotope effect we meant the change in N2O isotopic signature, as you suggest. To avoid confusion this will be changed in the manuscript to: 'the increase in $\delta^{18}O$, $\delta^{15}N$ and $\delta^{15}N^{SP}$ of the residual $N_2O$ due to $N_2O$ reduction, is related to $r_{N2O}$'

*L77-78: I don't understand "N2O mixing ratio between two N2O production pathways and rN2O".*

This will be changed to: '$N_2O$ mixing proportions of two $N_2O$ production pathways and $r_{N2O}$'

*Section 2.1: has this experiment been conducted in 2015/2016. Please confirm if this is the procedure that has been applied in the past and state clearly that this paper reports on past campaigns and current incubations of the same soil.*

Yes, the field campaigns were conducted in 2015 and 2016, and the incubations in 2018. This information will be added: 'The soil for incubations, upper 30cm layer, was collected on the

18.01.2018 from the experimental plot used previously for field campaigns and the incubation was conducted from 19.02.2018 to 05.03.2018.'

*Materials and Methods*
*L192 cf: On the one hand, water was added to obtain a certain water content in the soil. On the other hand, columns were continuously flushed with He/O2/N2. These gases are dry (i.e., very low dew points), so that they will take up significant amounts of water from the soil, considering a flush time of 5 days. As a consequence, the soil columns will dry out under this treatment. How did the authors account for this drying effect when referring to the WFPS values? Are the WFPS values an average or the initially values that was aimed at?*

The WFPS was analysed with mineral nitrogen analyses and do not show significant decrease. The given WFPS values are the actual measured values, not the theoretical target values. The results are presented in the supplement (Fig.S1). Water was also added in the middle of the experiment therefore the drying effect was not significant. The reference to the WFPS results shown in supplement will be added in this section: 'The WFPS values were controlled during the experiment (Fig. S1).'

*L210: I suggest mentioning that fluxes were calculated based on the dynamic chamber principle. Correction for the inlet concentration is omitted since the gas mixture used for flushing is N2O-free. Do I get this right?*

Yes, thank you, this will be added.

*L310: Please give a rationale for expecting higher fungal denitrification than Ni*

This is rather high soil moisture (>60% WFPS) and low ammonium content (Table 1). This explanation will be added in the text.

*L319: I suggest explaining what the cases are: Case 1 assumes reduction of N2O produced through bacterial denitrification and subsequent mixing, whereas case 2 assumes mixing and subsequent reduction of the mixed pool by denitrifiers. : : : if I am getting you right.*

Yes, this is right, but this is just explained a few lines above. I would like to rather avoid repetitions.

*L320: please explain c, and explain what happens if the condition is met.*

Sorry, this was a mistake in properly displaying of this formula, it should be:
$L_{i+1}/L_i \geq \alpha$,
'c' was a typo.
The condition of $L_{i+1}/L_i \geq \alpha$ is the main concept of standard Metropolis algorithm. The newly generated state is accepted as a valid configuration if the likelihood function ratio $L_{i+1}/L_i$ is greater or equal than a random variable $\alpha$. If the state is accepted it becomes a new

reference (appearing in the denominator) in the next iteration. If it is not accepted the next iteration is performed with unchanged value in the denominator in likelihood ratios.

*Results*
*L421-425: Actually, I can't follow this short statement entirely. I understand that the authors aim at getting rid of the implicit assumption of DELTAdelta maps that the dominant source of N2O is denitrification. I thank the authors for this advancement.*

There was one mistake in this paragraph: wrong word 'precursor' was used instead of 'endmember' (L422), which probably made the understanding of this section difficult. We will correct this and further clarify the points below.

*Some unclarities remain however:*
*1. For a 18O/SP map, the endmember values for bD, nD and fD are changed by the same amount, -6.4 per mil of water. Correcting for 18O of water makes sense, if nitrate exchanges O with water completely. However, when looking in the literature cited, it seems like the endmember values in the literature have been corrected for 18O already. For this reason, and I want to discuss this with the authors, I suggest correcting for the difference in d18O water between the presented study and the body of literature. Please clarify.*

Since the O-exchange is usually high correcting with water makes more sense that with nitrate. Of course, if we knew the exact O-exchange this could be done more precisely, but usually it is not known. These effects and associated uncertainties are discussed in Section 4.1, L641-655. The cited literature values (Table S1) are the isotope effects – so, the isotope shift between the water applied and the emitted N2O – to avoid confusion we will denote this with $\varepsilon$:

$\varepsilon_{N2O/H2O} = \delta_{N2O} - \delta_{H2O}$, e.g. $\delta_{N2O} = 10$, $\delta_{H2O} = -9$ => $\varepsilon_{N2O/H2O} = 19$

We aim to calculate the expected N2O $\delta$ values emitted from the particular pathways for our case studies, with $\delta H2O=-6.4$ so need to calculate:

$\delta_{N2O} = \varepsilon_{N2O/H2O} + \delta_{H2O} = 12.6$

All this values are presented in Table S1 in the supplement. But now I realise these are very important data for the study so this table will be moved to the main manuscript text and placed below the calculation explanations.

This more exact explanation will be added in the manuscript as well.

*2. Again, for a 18O/SP map, this correction is equivalent with a correction of the measured N2O 18O signature with water 18O, as correcting all endmembers for the same value is mathematically the same as correcting the measured values.*

We are not correcting all the endmembers with water, because $\delta 18O$ for Ni depends on atmospheric oxygen (since this is quite stable we do not assume stable value for this source). You are right – for the case of bD and fD mixing for SP/O Map it doesn't make difference but it does for bD-Ni mixing and for all results of the 3DI model.

*3. For 15N/SP maps, this is actually a further development, since this allows correcting both Ni and bD, fD and nD endmembers, in contrast to only correcting measured values with one assumed precursor composition.*
*I am asking the authors to comment on this and make this clear in the text if they agree. Please also explain the sign of your correction. It interferes with the definition of the apparent isotope effect in your reference 4 of the supplementary material (Sutka 2006).*

Thank you, this statement will be added in the manuscript. But this works for both SP/N and SP/O Map - because $\delta 18O$ for Ni in this approach is not corrected with water but can be corrected to the atmospheric oxygen (if different than mean value, which may be the case in e.g. aquatic studies).

We will add the definition of $\varepsilon$ to make the signs and recalculation procedures clear (the literature data is recalculated according to our definition). But I couldn't find the definition of the apparent isotope effect in (Sutka et al., 2006) and any supplementary material to this paper.

After addition of clarifications this paragraph will expand and will not fit anymore to results section. It will be moved to Methods section 2.5, as:

For the graphical presentation of dual isotope plots for sampling points always $\delta^{18}O$ and $\delta^{15}N$ values of emitted $N_2O$ are plotted ($\delta^{18}O_{N2O}$, $\delta^{15}N_{N2O}$). But the precursors isotopic signatures ($\delta^{18}O_{H2O}$, $\delta^{15}N_{NO3-}$, $\delta^{15}N_{NH4+}$) are taken into account by respective correction of mixing endmembers isotopic ranges (see Table S1). The literature endmember ranges are given as isotope effects ($\varepsilon$) expressed in relation to particular precursor relevant for particular pathway, e.g. for $\delta^{18}O$ of bD the $\varepsilon_{N2O/H2O}$ is calculated by subtracting the precursor isotopic signature ($\delta_{H2O}$) from the measured $\delta_{N2O}$ values:

$$\varepsilon_{N2O/precursor} = \delta_{N2O} - \delta_{precursor} \tag{11}$$

e.g. for $\delta^{18}O$ of bD: $\delta_{N2O} = 10$, $\delta_{H2O} = -9$; $\varepsilon_{N2O/H2O} = 19$

Afterwards, the literature isotope effects are corrected with the actually measured precursor values determined for the particular study ($\delta_{actual\ precursor}$) to determine the characteristic isotopic signature of $N_2O$ emitted from the particular mixing endmember for this particular study conditions ($\delta_{N2O,\ endmember}$):

$$\delta_{N2O\_endmember} = \varepsilon_{N2O/precursor} + \delta_{actual\ precursor} \tag{12}$$

e.g. $\delta^{18}O$ of bD: $\varepsilon_{N2O/H2O} = 19$, $\delta_{actual\ H2O} = -6.4$, $\delta_{N2O\_bD} = 12.6$.

Hence, the endmember ranges represent the expected isotopic signatures of $N_2O$ originating from each mixing endmember for the particular case study characterised by specific precursor isotopic signatures. Such approach allows for presenting all data in the common isotopic

scales without presumption on the dominating pathway and dominating precursor. Hence, this new approach presented here is actually a further development of Maps, since this allows for correcting both Ni and bD, fD and nD endmembers with relevant distinct precursors, in contrast to only correcting measured values with one common assumed precursor isotopic signature. In previous papers, where $\delta^{18}O$ and $\delta^{15}N$ related to precursors ($\delta^{18}O_{N2O/H2O}$, $\delta^{15}N_{N2O/NO3}$) were plotted (Ibraim et al., 2019; Lewicka-Szczebak et al., 2017; Lewicka-Szczebak et al., 2016) it was assumed that denitrification must be the dominating $N_2O$ production pathway.

*L434: Please define minimum reduction line. I guess it is the dashed red line, but it hasn't been defined in the figure caption.*

This definition will be added, it is indeed the dashed line. Sorry for the missing information.

*L461: there is a large discrepancy of field and laboratory NH4 delta values. This observation should also be taken up in the discussion again.*

This is most probably due to differences in fertilizer addition techniques. In field studies the fertilizer solution was injected into the soil intact columns and in laboratory studies it was mixed and afterwards packed into the vessels. In both studies we observe a very fast decrease in ammonium content which is most probably due to its adsorption. The more detailed interpretation of 15N experimental results including Ntrace model is ongoing, but probably during mixing this adsorption process is more enhanced when compared to injection technique. But this is just a speculation so far. The discussion on this issue will be extended in the follow up paper, where we also include the 15N-NH4 treatment which was not presented here. This information will be added in the manuscript.

*L495: In my opinion, the pie charts are i) not well described and ii) consume more space than necessary. If the authors stick to the pie charts, I ask them to indicate that the share of N2 produced during N2O reduction is given as N2 in percent. Hatched superimposed patterns show the source processes for N2O that has been reduced to N2. All in all, my suggestion is showing 2 bar plots (case 1 and case 2) with categories bD (bdN2O, bdN2), nD, fD, Ni, N2.*

Thank you for the nice idea with bar plots. We hoped to nicely present the comparison between cases with the pie plots, but apparently did not succeed with this idea, we fully agree that bar plots are better for this aim. This will be changed and such new Figures will be included in the manuscript:

[Figure]

*Figure 5: figure 5 gives everything but an overview of the results. I suggest showing only one case for Laboratory or field and instead have the numbers intelligible, and annotate scales. Please only show significant correlations, and don't correlate font size with correlation strength.*

Ok, I see your points. I think we can fully move this Figure to the supplement. Since the number of points (individual iterations) is very high – ca. 1000 points – most correlations are significant, even showing low R values. These are also standard graphs for the isotope mixing models introduced by the trophic nets research and we wanted to keep the idea similar. Showing 2 graphs only will not be informative, and actually the needed information is referred in the text and all the graphs will be presented in the supplement.

*L569: why is p<0.1 considered as significant? I suggest rephrasing the sentence, proved sounds odd in terms of statistical inference. The p agreement between reference method and SP/O Map (p<0.1??) as well as Case 2 of 3DI model (p<0.05) was statistically significant.*

p<0.1 may be assumed as statistically significant, we have changed this in the statistical methods, Section 2.7. Although only p<0.05 is usually accepted as significant, here we think that the values obtained for SP/O Map are also important indication and should be accepted, with a clear statement that this statistical significance is weak (p values are shown). The sentence will be corrected.

*Discussion L718-720: could the authors please explain how the uncertainty was actually transferred in the model. In line 331-332, the authors state that ": : :uncertainties of the source's data ifs fed into the model through the variance in the calculation of unnormalized likelihood". However, I don't understand then the following sentence, that the prior distribution was assumed uninformative. Does this mean that for each endmember isotopic composition, the whole range of values was allowed? Please clarify how this actually works.*

The sentence 'the prior distribution was assumed uninformative' is misleading. It refers to starting values for the model – we do not assume any preference for any pathway, that's why flat Dirichlet distribution is applied. We will modify this sentence to avoid confusion.

We have applied the sources data with their uncertainty into the model, as defined in Tab S1.

*Conclusion*
*L863-864: I think this needs to be rephrased. The authors haven't presented a triple isotope plot, but rather present a likelihood function that allows for including three isotopic quantities and 4 associated isotopic endmembers.*

This sentence will be corrected to:

- Here we present for the first time the idea of applying a model based on three $N_2O$ isotopic signatures. We are convinced that this is a powerful step forward in development of $N_2O$ isotopocule methods to quantify especially $r_{N2O}$, but also estimate some mixing proportions of the four $N_2O$ pathways included in the model.

---

## Author Comment (AC2) · 18 Sep 2020

*Author's response on*

**"N₂O isotope approaches for source partitioning of N₂O production and estimation of N₂O reduction – validation with ₁₅N gas-flux method in laboratory and field studies" by Dominika Lewicka-Szczebak et al.**

**Review response for Anonymous referee #2**

- [1] *comments from referees*
- [2] authors response
- [3] authors changes in manuscript

Thank you very much for your positive evaluation of the manuscript and your critical comments which helped us to improve our work.

*Line 61: It would be helpful to provide the actual value of sensitivity increase here so that a direct comparison can be made.*

This is about 80-fold increase in sensitivity. This information will be added in the text.

*Line 68: 'budget' is a more appropriate word here rather than emission. It is probably much easier to measure N2O fluxes directly if emission is intended.*

Thank you, this will be changed.

*Line 76: Should diffusion of N2O be taken into account as one of the processes determining the final N2O isotopes? Either here or in discussion, why the fractionation factor of diffusion is not considered in this study should be briefly mentioned.*

We consider rather enzymatic processes than diffusion to be rate-limiting since enzymatic isotope fractionation is rather determining the apparent isotope effect. This has been more deeply discussed in our previous publications (Lewicka-Szczebak et al., 2014, 2015) and we will add this information here.

*Line 98: What was the non-identical treatment here? Suggest briefly describe to give the readers some idea on the treatment differences which should be avoided*

Different fertilizer application procedures: needle injection of fertilizer solution for $^{15}$N treatments and surface distribution of fertilizer in NA treatments, different sizes of $^{15}$N and NA microplots and chambers). This information will be added.

*Line 117: Suggest the authors indicate their reference method here*

This is $^{15}$N gas-flux method. This information will be added.

*Line 159: Should also briefly mention what is the same treatment strategy employed in this study.*

This is: identical fertilizer application procedure as fertilizer solution applied with needle injection technique, identical water and fertilizer addition and identical plots and chamber sizes. This information will be added.

*Line 178: Why 20 mg N/kg of soil in lab incubation compared to 10 mg N/kg soil in field fertilisation?*

This was wrongly described and will be corrected for: in both lab and field study total fertilization was 20 mg N per kg soil added as $NaNO_3$ (10 mg N) and $NH_4Cl$ (10 mg N)).

*Line 141: This is confusing. Why the dates for 'next field campaigns' do not correspond to what is written earlier?*

These are dates when the cylinders were reinstalled, this was done at least one month before the next filed campaign. This will be clarified.

*Line 217: How was N2O converted to N2? In-line conversion?*

Yes, in-line reduction, this information will be added.

*Line 298: nD is not included here. Why? Understand that the isotopic ranges are not very different between bD and nD but authors should briefly mention why this is not included here to avoid confusion. Also to show that the authors have considered the nD pathway as well.*

nD cannot be really separated with this approach from bD. It will be clarified that the bD fraction here can possibly include nD as well.

*Line 310: Why higher fungal DN compared to nitrification in this instance? The data in Table S2 can be used to briefly justify this?*

We deal with rather high soil moisture, mostly over 65% WFPS, and also ammonium content was low, which rather favours fD than Ni. This explanation will be added.

*Line 379: Some of the N2 fluxes are above the detection limit but the values written here are below the detection limit mentioned in line 373.*

Sorry, this was a mistake, it is from 23 to 304 g N-N2. This will be corrected. Thank you for careful reading!

*Line 421 - 423: Don't quite get what you mean here. Consider rephrasing. You mean corrected precursor ranges based on different fractionation factors?*

It will be clarified. In this sentence one word was incorrectly used – precursors instead of endmembers. Sorry for this mistake.

*Line 434: The minimum reduction line is not described in Fig. 1. The dotted lines and the mixing should be clearly described in the legend/caption.*

This explanation will be added to the Fig.1 caption: The soild lines (bD-fD mixing and mean reduction line) are main assumptions used in the calculation procedures for SP/O Map. The grey dashed line shows the alternative bD-Ni mixing line (calculations with this alternative scenario are also presented in the supplement Table S2). The red dashed line shows the minimum reduction line – for the case of minimal delta values of the bD endmember. And for Fig.3 caption: The dashed line shows the linear fit for all the points with its equation and statistics above.

*Line 461: There was relatively large discrepancy between lab and field NO3- and NH4+ values. In fact, the d15N-NH4+ is very heavy and the possible factors driving these values should be discussed.*

We comment this in the discussion, L680: This indicates that the ammonium pool was highly fractionated and nearly exhausted. This is most probably due to adsorption processes. But this is just a speculation so far. The discussion on this issue will be extended in the follow up paper, where we also include the 15N-NH4 treatment which was not presented here. This information will be added in the manuscript.

*Line 490: The authors mentioned that the high d15N-NH4+ has shifted the location of the nD and Ni in the end member mixing plot. What is the author comparing the shift to?*

Ccompared to cases when similar $\delta^{15}N_{NH4}$ and $\delta^{15}N_{NO3}$ values are determined or assumed – this will be clarified in the text.

*Line 551: Amplitude for 3D1 model, case 1 is not always lower than the reference – at the start and towards the end of sampling, the amplitude is higher than the reference method. Any explanation on why this is the case?*

I meant lower amplitude of the temporal changes, this will be clarified in the text. The uncertainty of each method mostly depend on the standard deviation of 4 repetitions of which each time sample consists.

*Line 670: I agree with the authors that recalculation of the literature mixing endmember values is important but my question is what fractionation factors should be considered when correcting these values and how to evaluate that these corrected ranges are justified?*

We can take the literature ranges for fractionation factors based on pure culture studies (we have presented the summarised values in Table S1, they are also summarised in the supplement to new perspective paper Yu et al., 2020 (https://onlinelibrary.wiley.com/doi/abs/10.1002/rcm.8858)) . These values can be also determined experimentally for the particular soil under study, at least for denitrification, but this is complex and time consuming. But importantly the literature fractionation factors for particular processes must be corrected with the substrate isotopic signatures, which should be determined for each soil study. The procedure of this correction is presented in Table S1. We have also extended the description of this correction and will move this whole paragraph to the methods section.

For the graphical presentation of dual isotope plots for sampling points always $\delta^{18}O$ and $\delta^{15}N$ values of emitted $N_2O$ are plotted ($\delta^{18}O_{N2O}$, $\delta^{15}N_{N2O}$). But the precursors isotopic signatures ($\delta^{18}O_{H2O}$, $\delta^{15}N_{NO3-}$, $\delta^{15}N_{NH4+}$) are taken into account by respective correction of mixing endmembers isotopic ranges (see Table S1). The literature endmember ranges are given as isotope effects ($\epsilon$) expressed in relation to particular precursor relevant for particular pathway, e.g. for $\delta^{18}O$ of bD the $\epsilon_{N2O/H2O}$ is calculated by subtracting the precursor isotopic signature ($\delta_{H2O}$) from the measured $\delta_{N2O}$ values:

$$\epsilon_{N2O/precursor} = \delta_{N2O} - \delta_{precursor} \qquad (11)$$

e.g. for $\delta^{18}O$ of bD: $\delta_{N2O} = 10$, $\delta_{H2O} = -9$; $\epsilon_{N2O/H2O} = 19$

Afterwards, the literature isotope effects are corrected with the actually measured precursor values determined for the particular study ($\delta_{\text{actual precursor}}$) to determine the characteristic isotopic signature of $N_2O$ emitted from the particular mixing endmember for this particular study conditions ($\delta_{\text{N2O, endmember}}$):

$$\delta_{\text{N2O\_endmember}} = \varepsilon_{\text{N2O/precursor}} + \delta_{\text{actual precursor}} \tag{12}$$

e.g. $\delta^{18}O$ of bD: $\varepsilon_{\text{N2O/H2O}} = 19$, $\delta_{\text{actual H2O}} = -6.4$, $\delta_{\text{N2O\_bD}} = 12.6$.

Hence, the endmember ranges represent the expected isotopic signatures of $N_2O$ originating from each mixing endmember for the particular case study characterised by specific precursor isotopic signatures. Such approach allows for presenting all data in the common isotopic scales without presumption on the dominating pathway and dominating precursor. Hence, this new approach presented here is actually a further development of Maps, since this allows for correcting both Ni and bD, fD and nD endmembers with relevant distinct precursors, in contrast to only correcting measured values with one common assumed precursor isotopic signature. In previous papers, where $\delta^{18}O$ and $\delta^{15}N$ related to precursors ($\delta^{18}O_{\text{N2O/H2O}}$, $\delta^{15}N_{\text{N2O/NO3}}$) were plotted (Ibraim et al., 2019; Lewicka-Szczebak et al., 2017; Lewicka-Szczebak et al., 2016) it was assumed that denitrification must be the dominating $N_2O$ production pathway.

We will also move the Table 1 into the main manuscript, since it contains important information for these corrections.

*Line 686: Be specific of what shift is meant here? Temporal?*

Yes, temporal shift, this will be added.

*Line 820: This sentence is rather subjective. Is it possible to provide a more definitive range here? Can the authors make use of a sensitivity analysis to show the extent of substrate isotopic variations effects on the accuracy of the mapping approach?*

This is quite a complex analysis – it has been done for SP/O Map (Wu et al., 2019 https://www.sciencedirect.com/science/article/abs/pii/S0013935119306036) but not yet for isotope Maps applying d15N. This is definitely the topic for the further work and it is planned to be done soon. Without a precise analysis it is not possible to provide a precise numbers here.

*Line 825: Can the author suggest the lowest N2O fluxes without compromising the precision of isotope maps and the 2DI model? This will be helpful as a guideline for future studies wanting to use these approaches.*

Based on our F1 and F2 field case studies we can say that where $N_2O$ flux was mostly below 10 gN-$N_2$O ha$^{-1}$d$^{-1}$ the pathways partitioning was biased. This information will be added in the text.

---

## Author Response (AR2)

*Author's response on*

**"N₂O isotope approaches for source partitioning of N₂O production and estimation of N₂O reduction – validation with ₁₅N gas-flux method in laboratory and field studies" by Dominika Lewicka-Szczebak et al.**

**Review response for Anonymous referee #1**

- *(1)* *comments from referees*
- *(2)* authors response
- *(3)* authors changes in manuscript

Thank you very much for your positive evaluation of the manuscript and your critical comments which helped us to improve our work.

We will take into account all the comments and especially clarify the endmembers isotope values corrections we applied. Here we provide detailed responses for the raised points, minor corrections have been accepted and have been included into reviewed manuscript.

*L70 I am not sure if I understand correctly. The observed isotope effect for N2O reduction is quite stable and doesn't depend on r N2O. Do you mean "change in N2O isotopic composition due to N2O reduction depends largely on y n2o"?*

Isotope fractionation during N2O reduction is independent of rN2O (mostly). But with the isotope effect we meant the change in N2O isotopic signature, as you suggest. To avoid confusion this was changed in the manuscript to: 'the increase in $\delta^{18}O$, $\delta^{15}N$ and $\delta^{15}N^{SP}$ of the residual N₂O due to N₂O reduction, is related to $r_{N2O}$'

*L77-78: I don't understand "N2O mixing ratio between two N2O production pathways and rN2O".*

This was changed to: 'N₂O mixing proportions of two N₂O production pathways and $r_{N2O}$'

*Section 2.1: has this experiment been conducted in 2015/2016. Please confirm if this is the procedure that has been applied in the past and state clearly that this paper reports on past campaigns and current incubations of the same soil.*

Yes, the field campaigns were conducted in 2015 and 2016, and the incubations in 2018. This information was added: 'The soil for incubations, upper 30cm layer, was collected on the

18.01.2018 from the experimental plot used previously for field campaigns and the incubation was conducted from 19.02.2018 to 05.03.2018.'

*Materials and Methods*
*L192 cf: On the one hand, water was added to obtain a certain water content in the soil. On the other hand, columns were continuously flushed with He/O2/N2. These gases are dry (i.e., very low dew points), so that they will take up significant amounts of water from the soil, considering a flush time of 5 days. As a consequence, the soil columns will dry out under this treatment. How did the authors account for this drying effect when referring to the WFPS values? Are the WFPS values an average or the initially values that was aimed at?*

The WFPS was analysed with mineral nitrogen analyses and do not show significant decrease. The given WFPS values are the actual measured values, not the theoretical target values. The results are presented in the supplement (Fig.S1). Water was also added in the middle of the experiment therefore the drying effect was not significant. The reference to the WFPS results shown in supplement was added in this section: 'The WFPS values were controlled during the experiment (Fig. S1).'

*L210: I suggest mentioning that fluxes were calculated based on the dynamic chamber principle. Correction for the inlet concentration is omitted since the gas mixture used for flushing is N2O-free. Do I get this right?*

Yes, thank you, this was added: 'For laboratory incubations fluxes were calculated based on the dynamic chamber principle. Correction for the inlet concentration is omitted since the $N_2O$-free gas mixture was used for flushing:'

*L310: Please give a rationale for expecting higher fungal denitrification than Ni*

This is rather high soil moisture (>60% WFPS) and low ammonium content (Table 1). This explanation was added in the text: 'For our case studies, due to rather high soil moisture (>60% WFPS) and low ammonium content (Table 2), we rather expect higher fD contribution than Ni,'

*L319: I suggest explaining what the cases are: Case 1 assumes reduction of N2O produced through bacterial denitrification and subsequent mixing, whereas case 2 assumes mixing and subsequent reduction of the mixed pool by denitrifiers. : : : if I am getting you right.*

Yes, this is right, but this is just explained a few lines above. I would like to rather avoid repetitions.

*L320: please explain c, and explain what happens if the condition is met.*

Sorry, this was a mistake in properly displaying of this formula, it should be:
$L_{i+1}/L_i \geq \alpha$,

'c' was a typo.

The condition of $L_{i+1}/L_i \geq \alpha$ is the main concept of standard Metropolis algorithm. The newly generated state is accepted as a valid configuration if the likelihood function ratio $L_{i+1}/L_i$ is greater or equal than a random variable $\alpha$. If the state is accepted it becomes a new reference (appearing in the denominator) in the next iteration. If it is not accepted the next iteration is performed with unchanged value in the denominator in likelihood ratios.

*Results*
*L421-425: Actually, I can't follow this short statement entirely. I understand that the authors aim at getting rid of the implicit assumption of DELTAdelta maps that the dominant source of N2O is denitrification. I thank the authors for this advancement.*

There was one mistake in this paragraph: wrong word 'precursor' was used instead of 'endmember' (L422), which probably made the understanding of this section difficult. We have corrected this and further clarify the points below.

*Some unclarities remain however:*
*1. For a 18O/SP map, the endmember values for bD, nD and fD are changed by the same amount, -6.4 per mil of water. Correcting for 18O of water makes sense, if nitrate exchanges O with water completely. However, when looking in the literature cited, it seems like the endmember values in the literature have been corrected for 18O already. For this reason, and I want to discuss this with the authors, I suggest correcting for the difference in d18O water between the presented study and the body of literature. Please clarify.*

Since the O-exchange is usually high correcting with water makes more sense that with nitrate. Of course, if we knew the exact O-exchange this could be done more precisely, but usually it is not known. These effects and associated uncertainties are discussed in Section 4.1, L641-655. The cited literature values (Table S1) are the isotope effects – so, the isotope shift between the water applied and the emitted N2O – to avoid confusion we have denoted this with $\varepsilon$:

$\varepsilon_{N2O/H2O} = \delta_{N2O} - \delta_{H2O}$, e.g. $\delta_{N2O} = 10$, $\delta_{H2O} = -9$ => $\varepsilon_{N2O/H2O} = 19$

We aim to calculate the expected N2O $\delta$ values emitted from the particular pathways for our case studies, with $\delta H2O=-6.4$ so need to calculate:

$\delta_{N2O} = \varepsilon_{N2O/H2O} + \delta_{H2O} = 12.6$

All this values are presented in Table S1 in the supplement. But now I realise these are very important data for the study so this table will be moved to the main manuscript text and placed below the calculation explanations.

This more exact explanation was added in the manuscript as well.

*2. Again, for a 18O/SP map, this correction is equivalent with a correction of the measured N2O 18O signature with water 18O, as correcting all endmembers for the same value is mathematically the same as correcting the measured values.*

We are not correcting all the endmembers with water, because δ18O for Ni depends on atmospheric oxygen (since this is quite stable we do not assume stable value for this source). You are right – for the case of bD and fD mixing for SP/O Map it doesn't make difference but it does for bD-Ni mixing and for all results of the 3DI model.

*3. For 15N/SP maps, this is actually a further development, since this allows correcting both Ni and bD, fD and nD endmembers, in contrast to only correcting measured values with one assumed precursor composition.*
*I am asking the authors to comment on this and make this clear in the text if they agree. Please also explain the sign of your correction. It interferes with the definition of the apparent isotope effect in your reference 4 of the supplementary material (Sutka 2006).*

Thank you, this statement was added in the manuscript. But this works for both SP/N and SP/O Map - because δ18O for Ni in this approach is not corrected with water but can be corrected to the atmospheric oxygen (if different than mean value, which may be the case in e.g. aquatic studies).

We added the definition of $\varepsilon$ to make the signs and recalculation procedures clear (the literature data is recalculated according to our definition). But I couldn't find the definition of the apparent isotope effect in (Sutka et al., 2006) and any supplementary material to this paper.

After addition of clarifications this paragraph has expanded and do not fit anymore to results section. It was be moved to Methods section 2.5, as follows:

For the graphical presentation of dual isotope plots for sampling points always $\delta^{18}O$ and $\delta^{15}N$ values of emitted $N_2O$ are plotted ($\delta^{18}O_{N2O}$, $\delta^{15}N_{N2O}$). But the precursors isotopic signatures ($\delta^{18}O_{H2O}$, $\delta^{15}N_{NO3-}$, $\delta^{15}N_{NH4+}$) are taken into account by respective correction of mixing endmembers isotopic ranges (see Table S1). The literature endmember ranges are given as isotope effects ($\varepsilon$) expressed in relation to particular precursor relevant for particular pathway, e.g. for $\delta^{18}O$ of bD the $\varepsilon_{N2O/H2O}$ is calculated by subtracting the precursor isotopic signature ($\delta_{H2O}$) from the measured $\delta_{N2O}$ values:

$$\varepsilon_{N2O/precursor} = \delta_{N2O} - \delta_{precursor} \tag{11}$$

e.g. for $\delta^{18}O$ of bD: $\delta_{N2O} = 10$, $\delta_{H2O} = -9$; $\varepsilon_{N2O/H2O} = 19$

Afterwards, the literature isotope effects are corrected with the actually measured precursor values determined for the particular study ($\delta_{actual\ precursor}$) to determine the characteristic isotopic signature of $N_2O$ emitted from the particular mixing endmember for this particular study conditions ($\delta_{N2O,\ endmember}$):

$$\delta_{N2O\_endmember} = \varepsilon_{N2O/precursor} + \delta_{actual\ precursor} \tag{12}$$

e.g. $\delta^{18}O$ of bD: $\varepsilon_{N2O/H2O} = 19$, $\delta_{actual\ H2O} = -6.4$, $\delta_{N2O\_bD} = 12.6$.

Hence, the endmember ranges represent the expected isotopic signatures of $N_2O$ originating from each mixing endmember for the particular case study characterised by specific precursor isotopic signatures. Such approach allows for presenting all data in the common isotopic scales without presumption on the dominating pathway and dominating precursor. Hence, this new approach presented here is actually a further development of Maps, since this allows for correcting both Ni and bD, fD and nD endmembers with relevant distinct precursors, in contrast to only correcting measured values with one common assumed precursor isotopic signature. In previous papers, where $\delta^{18}O$ and $\delta^{15}N$ related to precursors ($\delta^{18}O_{N2O/H2O}$, $\delta^{15}N_{N2O/NO3}$) were plotted (Ibraim et al., 2019; Lewicka-Szczebak et al., 2017; Lewicka-Szczebak et al., 2016) it was assumed that denitrification must be the dominating $N_2O$ production pathway.

*L434: Please define minimum reduction line. I guess it is the dashed red line, but it hasn't been defined in the figure caption.*

Sorry for the missing information. This definition was added, it is indeed the dashed line: 'Figure 1: $N_2O$ isotope data of field (A, green points) and laboratory studies (B, purple points) in SP/O Map presented with literature endmember values and theoretical mixing (grey line) and reduction (red line) lines. The soild lines (bD-fD mixing and mean reduction line) are main assumptions used in the calculation procedures for SP/O Map. The grey dashed line shows the alternative bD-Ni mixing line (calculations with this alternative scenario are also presented in the supplement Table S1). The red dashed line shows the minimum reduction line – for the case of minimal delta values of the bD endmember.'

*L461: there is a large discrepancy of field and laboratory NH4 delta values. This observation should also be taken up in the discussion again.*

This is most probably due to differences in fertilizer addition techniques. In field studies the fertilizer solution was injected into the soil intact columns and in laboratory studies it was mixed and afterwards packed into the vessels. In both studies we observe a very fast decrease in ammonium content which is most probably due to its adsorption. The more detailed interpretation of 15N experimental results including Ntrace model is ongoing, but probably during mixing this adsorption process is more enhanced when compared to injection technique. But this is just a speculation so far. The discussion on this issue will be extended in the follow up paper, where we also include the 15N-NH4 treatment which was not presented here. This information was added in the manuscript (L745): 'This indicates that the ammonium pool was highly fractionated and nearly exhausted. This fast ammonium consumption will be further investigated in the follow up paper applying Ntrace model, where we also apply the $^{15}NH_4$ treatment for its proper interpretation (Müller et al., 2014).'

*L495: In my opinion, the pie charts are i) not well described and ii) consume more space than necessary. If the authors stick to the pie charts, I ask them to indicate that the share of N2 produced during N2O reduction is given as N2 in percent. Hatched superimposed patterns show the source processes for N2O that has been reduced to*

*N2. All in all, my suggestion is showing 2 bar plots (case 1 and case 2) with categories bD (bdN2O, bdN2), nD, fD, Ni, N2.*

Thank you for the nice idea with bar plots. We hoped to nicely present the comparison between cases with the pie plots, but apparently did not succeed with this idea, we fully agree that bar plots are better for this aim. This was changed and such new Figures was included in the manuscript:

[Figure]

*Figure 5: figure 5 gives everything but an overview of the results. I suggest showing only one case for Laboratory or field and instead have the numbers intelligible, and annotate scales. Please only show significant correlations, and don't correlate font size with correlation strength.*

Ok, I see your points. I think we can fully move this Figure to the supplement. Since the number of points (individual iterations) is very high – ca. 1000 points – most correlations are significant, even showing low R values. These are also standard graphs for the isotope mixing models introduced by the trophic nets research and we wanted to keep the idea similar. Showing 2 graphs only will not be informative, and actually the needed information is referred in the text and all the graphs are presented in the supplement.

*L569: why is p<0.1 considered as significant? I suggest rephrasing the sentence, proved sounds odd in terms of statistical inference. The p agreement between reference method and SP/O Map (p<0.1??) as well as Case 2 of 3DI model (p<0.05) was statistically significant.*

p<0.1 may be assumed as statistically significant, we have changed this in the statistical methods, Section 2.7. Although only p<0.05 is usually accepted as significant, here we think that the values obtained for SP/O Map are also important indication and should be accepted, with a clear statement that this statistical significance is weak (p values are shown).
The sentence was corrected: 'The statistically significant agreement was indicated for SP/O Map (p<0.1) and Case 2 of 3DI model (p<0.05)'

*Discussion L718-720: could the authors please explain how the uncertainty was actually transferred in the model. In line 331-332, the authors state that ": : :uncertainties of*

*the source's data ifs fed into the model through the variance in the calculation of unnormalized likelihood". However, I don't understand then the following sentence, that the prior distribution was assumed uninformative. Does this mean that for each endmember isotopic composition, the whole range of values was allowed? Please clarify how this actually works.*

The sentence 'the prior distribution was assumed uninformative' was misleading. It refers to starting values for the model – we do not assume any preference for any pathway, that's why flat Dirichlet distribution is applied. We modified this sentence in the methods section to avoid confusion (L385):

'For prior distributions of parameters flat Dirichlet distribution was used for proportional source contributions $f_i$ and uniform distribution for reduction parameter $r$.'
We have applied the sources data with their uncertainty into the model, as defined in Tab 1 – we also moved Table S1 to the main manuscript text.

*Conclusion*
*L863-864: I think this needs to be rephrased. The authors haven't presented a triple isotope plot, but rather present a likelihood function that allows for including three isotopic quantities and 4 associated isotopic endmembers.*

This sentence was corrected to:

- Here we present for the first time the idea of applying a model based on three $N_2O$ isotopic signatures. We are convinced that this is a powerful step forward in development of $N_2O$ isotopocule methods to quantify especially $r_{N2O}$, but also estimate some mixing proportions of the four $N_2O$ pathways included in the model.

**Review response for Anonymous referee #2**

(1) *comments from referees*
(2) *authors response*
(3) authors changes in manuscript

Thank you very much for your positive evaluation of the manuscript and your critical comments which helped us to improve our work.

*Line 61: It would be helpful to provide the actual value of sensitivity increase here so that a direct comparison can be made.*

This is about 80-fold increase in sensitivity. This information was added in the text: 'This new approach increases the sensitivity of $^{15}N$ gas-flux method (80-fold better sensitivity for $N_2+N_2O$ flux measurements (Well et al., 2019a))'

*Line 68: 'budget' is a more appropriate word here rather than emission. It is probably much easier to measure N2O fluxes directly if emission is intended.*

Thank you, this was changed.

*Line 76: Should diffusion of N2O be taken into account as one of the processes determining the final N2O isotopes? Either here or in discussion, why the fractionation factor of diffusion is not considered in this study should be briefly mentioned.*

We consider rather enzymatic processes than diffusion to be rate-limiting since enzymatic isotope fractionation is rather determining the apparent isotope effect. This has been more deeply discussed in our previous publications (Lewicka-Szczebak et al., 2014, 2015) and we added this information here: 'But on the other hand, complexity of the $N_2O$ production pathways with co-occurring $N_2O$ reduction, variability of isotope effects and isotope fractionation associated with diffusion processes can make this estimation imprecise (Lewicka-Szczebak et al., 2015; Lewicka-Szczebak et al., 2014; Yu et al., 2020).'

*Line 98: What was the non-identical treatment here? Suggest briefly describe to give the readers some idea on the treatment differences which should be avoided*

Different fertilizer application procedures: needle injection of fertilizer solution for $^{15}N$ treatments and surface distribution of fertilizer in NA treatments, different sizes of $^{15}N$ and NA microplots and chambers). This information was added:

'Due to non-identical treatment (different fertilizer application procedures: needle injection of fertilizer solution for $^{15}$N treatments and surface distribution of fertilizer in NA treatments; different sizes of $^{15}$N and NA microplots and chambers) '

*Line 117: Suggest the authors indicate their reference method here*

This is $^{15}$N gas-flux method. This information was added:

'by parallel application with the reference method - $^{15}$N gas-flux method.'

*Line 159: Should also briefly mention what is the same treatment strategy employed in this study.*

This is: identical fertilizer application procedure as fertilizer solution applied with needle injection technique, identical water and fertilizer addition  and identical plots and chamber sizes. This information was added:

'experiments applying an identical treatment strategy (meaning identical fertilizer application procedure: fertilizer solution applied with needle injection technique, identical water and fertilizer addition and identical plots and chamber sizes).'

*Line 178: Why 20 mg N/kg of soil in lab incubation compared to 10 mg N/kg soil in field fertilisation?*

This was wrongly described and was corrected for:

'in both lab and field study total fertilization was  20 mg N per kg soil added as NaNO$_3$ (10 mg N) and NH$_4$Cl (10 mg N)).'

*Line 141: This is confusing. Why the dates for 'next field campaigns' do not correspond to what is written earlier?*

These are dates when the cylinders were reinstalled, this was done at least one month before the next filed campaign. This was clarified:

'After each field campaign the cylinders were removed, cleaned and later reinstalled on new locations (on 27 Nov 2015 for F2 sampling and on 28 April 2016 for F3 sampling) for the next field campaign.'

*Line 217: How was N2O converted to N2? In-line conversion?*

Yes, in-line reduction, this information was added:

'In this set-up, $N_2O$ is converted to $N_2$ during in-line reduction, and stable isotope ratios $^{29}R$ $(^{29}N_2/^{28}N_2)$ and $^{30}R$ $(^{30}N_2/^{29}N_2)$, of $N_2$, of the sum of denitrification products $(N_2+N_2O)$ and of $N_2O$ are determined.'

*Line 298: nD is not included here. Why? Understand that the isotopic ranges are not very different between bD and nD but authors should briefly mention why this is not included here to avoid confusion. Also to show that the authors have considered the nD pathway as well.*

nD cannot be really separated with this approach from bD. It was clarified that the bD fraction here can possibly include nD as well:

' The Mapping approach is based on the different slopes of the mixing line between bD (possibly including also nD) and fD or Ni and the reduction line'

*Line 310: Why higher fungal DN compared to nitrification in this instance? The data in Table S2 can be used to briefly justify this?*

We deal with rather high soil moisture, mostly over 65% WFPS, and also ammonium content was low, which rather favours fD than Ni. This explanation was added:

'For our case studies, due to rather high soil moisture (>60% WFPS) and low ammonium content (Table 2), we rather expect higher fD contribution than Ni'

*Line 379: Some of the N2 fluxes are above the detection limit but the values written here are below the detection limit mentioned in line 373.*

Sorry, this was a mistake, it is from 23 to 304 g N-N2. This was corrected. Thank you for careful reading!

'varied from 23 to 304 g N-$N_2$ ha$^{-1}$ d$^{-1}$.'

*Line 421 - 423: Don't quite get what you mean here. Consider rephrasing. You mean corrected precursor ranges based on different fractionation factors?*

It was clarified (L334-336). In this sentence one word was incorrectly used – precursors instead of endmembers. Sorry for this mistake.

'For the graphical presentation of dual isotope plots for sampling points always $\delta^{18}$O and $\delta^{15}$N values of emitted $N_2O$ are plotted ($\delta^{18}O_{N2O}$, $\delta^{15}N_{N2O}$). But the precursors isotopic signatures ($\delta^{18}O_{H2O}$, $\delta^{15}N_{NO3-}$, $\delta^{15}N_{NH4+}$) are taken into account by respective correction of mixing endmembers isotopic ranges (see Table 1).'

*Line 434: The minimum reduction line is not described in Fig. 1. The dotted lines and the mixing should be clearly described in the legend/caption.*

This explanation will be added to the Fig.1 caption:

'The soild lines (bD-fD mixing and mean reduction line) are main assumptions used in the calculation procedures for SP/O Map. The grey dashed line shows the alternative bD-Ni mixing line (calculations with this alternative scenario are also presented in the supplement Table S2). The red dashed line shows the minimum reduction line – for the case of minimal delta values of the bD endmember'.

And for Fig.3 caption:

'The dashed line shows the linear fit for all the points with its equation and statistics above.'

*Line 461: There was relatively large discrepancy between lab and field NO3- and NH4+ values. In fact, the d15N-NH4+ is very heavy and the possible factors driving these values should be discussed.*

We comment this in the discussion, L745: This indicates that the ammonium pool was highly fractionated and nearly exhausted. This is most probably due to adsorption processes. But this is just a speculation so far. The discussion on this issue will be extended in the follow up paper, where we also include the 15N-NH4 treatment which was not presented here. This information was added in the manuscript:

'This indicates that the ammonium pool was highly fractionated and nearly exhausted. This fast ammonium consumption will be further investigated in the follow up paper applying Ntrace model, where we also apply the $^{15}NH_4$ treatment for its proper interpretation (Müller et al., 2014).'

*Line 490: The authors mentioned that the high d15N-NH4+ has shifted the location of the nD and Ni in the end member mixing plot. What is the author comparing the shift to?*

Compared to cases when similar $\delta^{15}N_{NH4}$ and $\delta^{15}N_{NO3}$ values are determined or assumed – this was clarified in the text:

'The application of Maps applying $\delta^{15}$N data, *i.e.*, SP/N and O/N Map, is very imprecise for this case study due to untypically high $\delta^{15}N_{NH4}$ values and shifted location of the nD and Ni

mixing endmembers (Fig. 2, Fig. 3) when compared to cases when similar $\delta^{15}N_{NH4}$ and $\delta^{15}N_{NO3}$ values are determined or assumed.'

*Line 551: Amplitude for 3D1 model, case 1 is not always lower than the reference – at the start and towards the end of sampling, the amplitude is higher than the reference method. Any explanation on why this is the case?*

I meant lower amplitude of the temporal changes, this was clarified in the text. The uncertainty of each method mostly depend on the standard deviation of 4 repetitions of which each time sample consists.

'All three estimates present the same trend as the reference method, however, with lower amplitude of the temporal change (Fig. 5B).'

*Line 670: I agree with the authors that recalculation of the literature mixing endmember values is important but my question is what fractionation factors should be considered when correcting these values and how to evaluate that these corrected ranges are justified?*

We can take the literature ranges for fractionation factors based on pure culture studies (we have presented the summarised values in Table S1 (Table 1 in the revised manuscript), they are also summarised in the supplement to new perspective paper Yu et al., 2020 (https://onlinelibrary.wiley.com/doi/abs/10.1002/rcm.8858)) . These values can be also determined experimentally for the particular soil under study, at least for denitrification, but this is complex and time consuming. But importantly the literature fractionation factors for particular processes must be corrected with the substrate isotopic signatures, which should be determined for each soil study. The procedure of this correction is presented in Table 1. We have also extended the description of this correction and moved this whole paragraph to the methods section, as follows:

'For the graphical presentation of dual isotope plots for sampling points always $\delta^{18}O$ and $\delta^{15}N$ values of emitted $N_2O$ are plotted ($\delta^{18}O_{N2O}$, $\delta^{15}N_{N2O}$). But the precursors isotopic signatures ($\delta^{18}O_{H2O}$, $\delta^{15}N_{NO3-}$, $\delta^{15}N_{NH4+}$) are taken into account by respective correction of mixing endmembers isotopic ranges (see Table S1). The literature endmember ranges are given as isotope effects ($\varepsilon$) expressed in relation to particular precursor relevant for particular pathway, e.g. for $\delta^{18}O$ of bD the $\varepsilon_{N2O/H2O}$ is calculated by subtracting the precursor isotopic signature ($\delta_{H2O}$) from the measured $\delta_{N2O}$ values:

$$\varepsilon_{N2O/precursor} = \delta_{N2O} - \delta_{precursor} \tag{11}$$

e.g. for $\delta^{18}O$ of bD: $\delta_{N2O} = 10$, $\delta_{H2O} = -9$; $\varepsilon_{N2O/H2O} = 19$

Afterwards, the literature isotope effects are corrected with the actually measured precursor values determined for the particular study ($\delta_{actual\ precursor}$) to determine the characteristic isotopic signature of $N_2O$ emitted from the particular mixing endmember for this particular study conditions ($\delta_{N2O,\ endmember}$):

$$\delta_{N2O\_endmember} = \varepsilon_{N2O/precursor} + \delta_{actual\ precursor} \tag{12}$$

e.g. $\delta^{18}O$ of bD: $\varepsilon_{N2O/H2O} = 19$, $\delta_{actual\ H2O} = -6.4$, $\delta_{N2O\_bD} = 12.6$.

Hence, the endmember ranges represent the expected isotopic signatures of $N_2O$ originating from each mixing endmember for the particular case study characterised by specific precursor isotopic signatures. Such approach allows for presenting all data in the common isotopic scales without presumption on the dominating pathway and dominating precursor. Hence, this new approach presented here is actually a further development of Maps, since this allows for correcting both Ni and bD, fD and nD endmembers with relevant distinct precursors, in contrast to only correcting measured values with one common assumed precursor isotopic signature. In previous papers, where $\delta^{18}O$ and $\delta^{15}N$ related to precursors ($\delta^{18}O_{N2O/H2O}$, $\delta^{15}N_{N2O/NO3}$) were plotted (Ibraim et al., 2019; Lewicka-Szczebak et al., 2017; Lewicka-Szczebak et al., 2016) it was assumed that denitrification must be the dominating $N_2O$ production pathway.'

We have also moved the Table 1 into the main manuscript, since it contains important information for these corrections.

*Line 686: Be specific of what shift is meant here? Temporal?*

Yes, temporal shift, this was added:

'However, it must be noted that in case of temporal shifts in the isotopic composition of the N or O substrate'

*Line 820: This sentence is rather subjective. Is it possible to provide a more definitive range here? Can the authors make use of a sensitivity analysis to show the extent of substrate isotopic variations effects on the accuracy of the mapping approach?*

This is quite a complex analysis – it has been done for SP/O Map (Wu et al., 2019 https://www.sciencedirect.com/science/article/abs/pii/S0013935119306036) but not yet for isotope Maps applying d15N. This is definitely the topic for the further work and it is planned to be done soon. Without a precise analysis it is not possible to provide a precise numbers here.

*Line 825: Can the author suggest the lowest N2O fluxes without compromising the precision of isotope maps and the 2DI model? This will be helpful as a guideline for future studies wanting to use these approaches.*

Based on our F1 and F2 field case studies we can say that where $N_2O$ flux was mostly below 10 gN-$N_2O$ ha$^{-1}$d$^{-1}$ the pathways partitioning was biased. This information was added in the text:

[revised manuscript text omitted]

Sformatowano … [1]
Sformatowana tabela … [2]
Sformatowano … [3]
Sformatowano … [4]
Sformatowano … [5]
Sformatowano … [7]
Sformatowano … [8]
Sformatowano … [6]
Sformatowano … [9]
Sformatowano … [10]
Sformatowano … [11]
Sformatowano … [12]
Sformatowano … [13]
Sformatowana tabela … [14]
Sformatowano … [15]
Sformatowano … [16]
Sformatowano … [17]
Sformatowano … [18]
Sformatowano … [20]
Sformatowano … [19]
Sformatowano … [21]
Sformatowano … [22]
Sformatowano … [23]
Sformatowano … [25]
Sformatowano … [27]
Sformatowano … [24]
Sformatowano … [26]
Sformatowano … [28]
Sformatowano … [29]
Sformatowano … [30]
Sformatowano … [31]
Sformatowano … [32]
Sformatowano … [33]
Sformatowano … [34]
Sformatowano … [35]
Sformatowano … [36]
Sformatowano … [37]
Sformatowano … [38]
Sformatowano … [39]
Sformatowano … [40]

[revised manuscript text omitted]

**Sformatowano:** Interlinia: 1,5 wiersza, Pozycja: Poziomo: 0,26 cm, Względem: Margines, Pionowo: 0,17 cm, Względem: Akapit, Poziomo: 0,25 cm, Zawijaj wokół

**Sformatowana tabela**

**Wstawione komórki**

**Sformatowano:** Niemiecki (Niemcy)

**Sformatowano:** Do lewej, Interlinia: 1,5 wiersza, Pozycja: Poziomo: 0,26 cm, Względem: Margines, Pionowo: 0,17 cm, Względem: Akapit, Poziomo: 0,25 cm, Zawijaj wokół

**Sformatowano:** Niemiecki (Niemcy)

**Sformatowano:** Wyjustowany, Interlinia: 1,5 wiersza, Pozycja: Poziomo: 0,26 cm, Względem: Margines, Pionowo: 0,17 cm, Względem: Akapit, Poziomo: 0,25 cm, Zawijaj wokół

[Figure]

**Sformatowano:** Niemiecki (Niemcy)

**Sformatowano:** Niemiecki (Niemcy)

F2 Case1

F2 Case2

F3 Case1

F3 Case2

L1 Case1

L1 Case2

[Figure]

L2 Case1

L2 Case2

Fig. 5 Matrix plots presenting detailed 3DI model outputs for each sampling date – here representative examples for each sampling campaign are shown (in the supplement plots for all samples are shown, Fig. S4). The plots in the diagonal show histograms of posterior probability distribution of $r_{N2O}$ and mixing ratios (scale from 0, left to 1, right), the plots above the diagonal show correlations between the modeled fractions (scale from 0, left to 1, right) and the values below the diagonal show $R$ coefficient of these correlations: in blue for positive correlations and in red for negative correlations with the size proportional to the $R$ value.

1305

[Figure]

 **Fig.6Figure 5:** Comparison of time changes in residual $N_2O$ fraction ($r_{N2O}$) determined with O/SP Map Case 1 and 3DI model with the reference method ($^{15}$N gas-flux). For the 3DI model results the 95% confidence interval is shown with grey shaded areas. Error bars for O/SP Map and $^{15}$N gas-flux data represent the standard deviation of replicate samples (n=4). For $N_2$ fluxes below the detection limit the estimated $r_{N2O}$ values are shown (red areas), calculated with $N_2$ flux from 0 to 1 of the detection limit.

[Figure]

**Sformatowano:** Czcionka: 11 pt, Niemiecki (Niemcy)

**Sformatowano:** Czcionka: 11 pt, Niemiecki (Niemcy)

**Sformatowano:** Czcionka: 11 pt, Niemiecki (Niemcy)

1315 Figure 6: **Comparison of 1:1 fit between $r_{N2O}$ determined with the reference method ([15]N gas-flux) and (A) 3DI model Case 2, (B) SP/O Map Case 1.**

[Figure]

**Sformatowano:** Czcionka: 11 pt

**Sformatowano:** Czcionka: 11 pt

**Sformatowano:** Czcionka: 11 pt, Niemiecki (Niemcy)

**Sformatowano:** Czcionka: 11 pt, Niemiecki (Niemcy)

**Sformatowano:** Czcionka: 11 pt, Niemiecki (Niemcy)

**Sformatowano:** Czcionka: 11 pt

**Sformatowano:** Czcionka: 11 pt

**Sformatowano:** Czcionka: 11 pt

**Sformatowano:** Czcionka: 11 pt, Niemiecki (Niemcy)

**Sformatowano:** Czcionka: 11 pt, Niemiecki (Niemcy)

**Sformatowano:** Czcionka: 11 pt, Niemiecki (Niemcy)

**Figure 7:** Comparison of $N_2O$ fractions comprising bacterial denitrification ($f_{bD}$) determined with O/SP Map Case 1 (representing bD+nD) and 3DI model Case 2 (respective fractions determined: bD, bD+nD, bD+fD) with the reference method ($^{15}$N gas-flux). $^{15}$N gas-flux method determines the $f_{P\_N2O}$ – $^{15}$N-pool derived fraction – comprising all $N_2O$ origins utilizing $^{15}$N-labelled $NO_3^-$ – theoretically mostly bD and fD. See Sections 4.2 and 4.3 for further discussion. For the 3DI model results the 95% confidence interval is shown with shaded areas. Error bars for O/SP Map and $^{15}$N gas-flux data represent the standard deviation of replicate samples (n=4).

| Strona 39: [1] Sformatowano | Dominika | 05.10.2020 10:40:00 |
|---|---|---|

Czcionka: 11 pt

| Strona 39: [2] Zmień | Dominika | 05.10.2020 10:40:00 |
|---|---|---|

Sformatowana tabela

| Strona 39: [3] Sformatowano | Dominika | 05.10.2020 10:40:00 |
|---|---|---|

Czcionka: 11 pt

| Strona 39: [4] Sformatowano | Dominika | 05.10.2020 10:40:00 |
|---|---|---|

Czcionka: 11 pt, Niemiecki (Niemcy)

| Strona 39: [5] Sformatowano | Dominika | 05.10.2020 10:40:00 |
|---|---|---|

Czcionka: 11 pt, Niemiecki (Niemcy)

| Strona 39: [6] Sformatowano | Dominika | 05.10.2020 10:40:00 |
|---|---|---|

Czcionka: 11 pt

| Strona 39: [6] Sformatowano | Dominika | 05.10.2020 10:40:00 |
|---|---|---|

Czcionka: 11 pt

| Strona 39: [6] Sformatowano | Dominika | 05.10.2020 10:40:00 |
|---|---|---|

Czcionka: 11 pt

| Strona 39: [6] Sformatowano | Dominika | 05.10.2020 10:40:00 |
|---|---|---|

Czcionka: 11 pt

| Strona 39: [7] Sformatowano | Dominika | 05.10.2020 10:40:00 |
|---|---|---|

Czcionka: 11 pt

| Strona 39: [8] Sformatowano | Dominika | 05.10.2020 10:40:00 |
|---|---|---|

Czcionka: 11 pt, Niemiecki (Niemcy)

| Strona 39: [9] Sformatowano | Dominika | 05.10.2020 10:40:00 |
|---|---|---|

Czcionka: 11 pt

| Strona 39: [10] Sformatowano | Dominika | 05.10.2020 10:40:00 |
|---|---|---|

Czcionka: 11 pt, Niemiecki (Niemcy)

| Strona 39: [10] Sformatowano | Dominika | 05.10.2020 10:40:00 |
|---|---|---|

Czcionka: 11 pt, Niemiecki (Niemcy)

| Strona 39: [11] Sformatowano | Dominika | 05.10.2020 10:40:00 |
|---|---|---|

Czcionka: 11 pt

| Strona 39: [12] Sformatowano | Dominika | 05.10.2020 10:40:00 |
|---|---|---|

Czcionka: 11 pt

| Strona 39: [13] Sformatowano | Dominika | 05.10.2020 10:40:00 |
|---|---|---|

Czcionka: 11 pt

| Strona 39: [14] Zmień | Dominika | 05.10.2020 10:40:00 |
|---|---|---|

Sformatowana tabela

| Strona 39: [15] Sformatowano | Dominika | 05.10.2020 10:40:00 |

Czcionka: 11 pt, Niemiecki (Niemcy)

| Strona 39: [15] Sformatowano | Dominika | 05.10.2020 10:40:00 |

Czcionka: 11 pt, Niemiecki (Niemcy)

| Strona 39: [15] Sformatowano | Dominika | 05.10.2020 10:40:00 |

Czcionka: 11 pt, Niemiecki (Niemcy)

| Strona 39: [16] Sformatowano | Dominika | 05.10.2020 10:40:00 |

Czcionka: 11 pt

| Strona 39: [17] Sformatowano | Dominika | 05.10.2020 10:40:00 |

Czcionka: 11 pt

| Strona 39: [18] Sformatowano | Dominika | 05.10.2020 10:40:00 |

Czcionka: 11 pt, Niemiecki (Niemcy)

| Strona 39: [19] Sformatowano | Dominika | 05.10.2020 10:40:00 |

Czcionka: 11 pt

| Strona 39: [20] Sformatowano | Dominika | 05.10.2020 10:40:00 |

Czcionka: 11 pt, Niemiecki (Niemcy)

| Strona 39: [21] Sformatowano | Dominika | 05.10.2020 10:40:00 |

Czcionka: 11 pt

| Strona 39: [22] Sformatowano | Dominika | 05.10.2020 10:40:00 |

Czcionka: 11 pt

| Strona 39: [23] Sformatowano | Dominika | 05.10.2020 10:40:00 |

Czcionka: 11 pt, Niemiecki (Niemcy)

| Strona 39: [24] Sformatowano | Dominika | 05.10.2020 10:40:00 |

Czcionka: 11 pt

| Strona 39: [25] Sformatowano | Dominika | 05.10.2020 10:40:00 |

Czcionka: 11 pt, Niemiecki (Niemcy)

| Strona 39: [26] Sformatowano | Dominika | 05.10.2020 10:40:00 |

Czcionka: 11 pt

| Strona 39: [27] Sformatowano | Dominika | 05.10.2020 10:40:00 |

Czcionka: 11 pt, Niemiecki (Niemcy)

| Strona 39: [28] Sformatowano | Dominika | 05.10.2020 10:40:00 |

Czcionka: 11 pt

| Strona 39: [28] Sformatowano | Dominika | 05.10.2020 10:40:00 |

Czcionka: 11 pt

| Strona 39: [28] Sformatowano | Dominika | 05.10.2020 10:40:00 |

Czcionka: 11 pt

| Strona 39: [29] Sformatowano | Dominika | 05.10.2020 10:40:00 |
|---|---|---|

Czcionka: 11 pt

| Strona 39: [30] Sformatowano | Dominika | 05.10.2020 10:40:00 |
|---|---|---|

Czcionka: 11 pt

| Strona 39: [31] Sformatowano | Dominika | 05.10.2020 10:40:00 |
|---|---|---|

Czcionka: 11 pt

| Strona 39: [32] Sformatowano | Dominika | 05.10.2020 10:40:00 |
|---|---|---|

Czcionka: 11 pt

| Strona 39: [33] Sformatowano | Dominika | 05.10.2020 10:40:00 |
|---|---|---|

Czcionka: 11 pt

| Strona 39: [34] Sformatowano | Dominika | 05.10.2020 10:40:00 |
|---|---|---|

Czcionka: 11 pt

| Strona 39: [35] Sformatowano | Dominika | 05.10.2020 10:40:00 |
|---|---|---|

Czcionka: 11 pt

| Strona 39: [36] Sformatowano | Dominika | 05.10.2020 10:40:00 |
|---|---|---|

Czcionka: 11 pt

| Strona 39: [37] Sformatowano | Dominika | 05.10.2020 10:40:00 |
|---|---|---|

Czcionka: 11 pt

| Strona 39: [38] Sformatowano | Dominika | 05.10.2020 10:40:00 |
|---|---|---|

Czcionka: 11 pt

| Strona 39: [39] Sformatowano | Dominika | 05.10.2020 10:40:00 |
|---|---|---|

Czcionka: 11 pt

| Strona 39: [40] Sformatowano | Dominika | 05.10.2020 10:40:00 |
|---|---|---|

Czcionka: 11 pt